# Learning Tractable Distributions of Language Model Continuations

## Abstract

Controlled language generation conditions text on sequence-level constraints (for example, syntax, style, or safety). These constraints may depend on future tokens, which makes directly conditioning an autoregressive language model (LM) generally intractable. Prior work uses tractable surrogates such as hidden Markov models (HMMs) to approximate the distribution over continuations and adjust the model's next-token logits at decoding time. However, we find that these surrogates are often weakly context aware, which reduces query quality. We propose *Learning to Look Ahead* (LTLA), a hybrid approach that pairs the same base language model for rich prefix encoding with a fixed tractable surrogate model that computes exact continuation probabilities. Two efficiency pitfalls arise when adding neural context: (i) naïvely rescoring the prefix with every candidate next token requires a sweep over the entire vocabulary at each step, and (ii) predicting fresh surrogate parameters for each prefix, although tractable at a single step, forces recomputation of future probabilities for every new prefix and eliminates reuse. LTLA avoids both by using a single batched HMM update to account for all next-token candidates at once, and by conditioning only the surrogate's latent state prior on the LM's hidden representations while keeping the surrogate decoder fixed, so computations can be reused across prefixes. Empirically, LTLA attains higher conditional likelihood than an unconditional HMM, approximates continuation distributions for vision–language models where a standalone HMM cannot encode visual context, and improves constraint satisfaction at comparable fluency on controlled-generation tasks, with minimal inference overhead.

## 1 Introduction

Autoregressive models are the dominant way to represent high-dimensional discrete distributions over language and sequence data (Grattafiori et al., 2024; Shin et al., 2021), factorizing the distribution into a sequence of next-token conditional distributions. Many useful queries, however, concern properties of the entire sequence: to generate autoregressively under a sequence-level constraint, we need the probability that the remaining suffix will satisfy that constraint given the current prefix (Boyd et al., 2022; Zhang et al., 2023). Computing this probability requires reasoning over exponentially many possible futures and is therefore intractable for standard autoregressive language models (LM). In practice, estimates come from sampling (Qin et al., 2022; Hu et al., 2023; Lew et al., 2023), which introduces significant computational overhead, or from learned heuristics (Krause et al., 2020; Yang & Klein, 2021; Meng et al., 2022) that must be tailored to each query or constraint.

Tractable probabilistic models (TPMs) are a broad class of generative models that admit efficient routines for exactly computing such conditional queries exactly and efficiently (Choi et al., 2020). As such, a natural approach employed by prior works (Zhang et al., 2023; Yidou-Weng et al., 2025) is to utilize TPMs as *tractable surrogates* to the LM, such that queries on the LM can be approximated by computing them on the surrogate. However, we identify that current TPMs used for language modeling, such as hidden Markov models (HMM), face two significant obstacles in this setting. Firstly, we find empirically that HMMs are comparatively insensitive to the prefix, which yields less accurate distributions over continuations, as illustrated in Fig. 1. Secondly, as models operating only over discrete sequences, they cannot effectively encode some kinds of context, notably continuous multimodal features used by modern vision–language models.

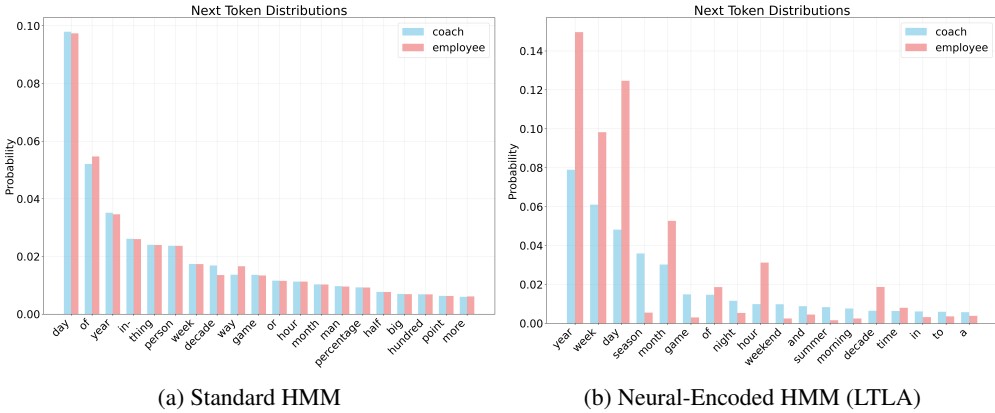

(a) Standard HMM            (b) Neural-Encoded HMM (LTLA)

Figure 1: The encoder given by standard HMMs is often insensitive to information contained within the context. In this example, we show an example with the context `they fired the <x> after just one`, where `<x>` can be `coach` or `employee`. The distribution is almost identical for the standard HMM, while the neural HMM shows a significant shift in distribution (in particular, with `season` and `game` being more likely when `<x>` = `coach`).

To encode the past well and answer future queries quickly, we separate the two jobs. Understanding the past ("lookback") should preserve as much information as possible, whereas forecasting the future ("lookahead") must remain tractable. The same base LM already builds a rich representation of the prefix, which we reuse to encode the prefix into the tractable model (for example, a HMM) that models and computes exact probabilities over continuations. In short, we propose a *hybrid* tractable model where a transformer-based LM is used for lookback and a HMM for lookahead.

Making this hybrid model practical raises two concrete challenges. First, when autoregressively generating the next token, a naïve strategy would run the transformer on the prefix concatenated with every candidate token across the vocabulary to assess how each affects the downstream conditional query; this per-step sweep over the entire vocabulary is expensive *(see "Exhaustive LM Rescoring" in Table 1)* and conceptually corresponds to a branching tree of continuations, as shown in Fig. 2(a). Second, conditioning the surrogate itself on the prefix by predicting fresh parameters for every context remains tractable at a single step but prevents reuse: as the prefix grows, the surrogate must be rebuilt and its future probabilities recomputed at every step, which increases decode-time cost and memory *(see "Prefix-Parameterized TPM" in Table 1; cf. Sec. 3.1)*. Both issues undercut the efficiency that makes TPMs attractive.

Thus, we propose a solution that encodes rich context with the transformer while maintaining TPM efficiency by encoding the prefix only into the HMM surrogate's latent prior and keeping the rest of the HMM fixed. Specifically, the transformer's hidden states are fed into a lightweight head that is trained to produce a prior over the surrogate's latent state, and the observed next token $x_t$ is incorporated with a one-step HMM update as a single batched matrix–vector operation, avoiding separate computation for each token in the vocabulary. With fixed transition and emission parameters, future computations are reusable across steps, so conditional queries remain exact and fast even for long sequence lengths *(see "LTLA (Ours)" in Table 1)*. Fig. 2(b) illustrates this design: the LM handles lookback, while a linear-chain HMM surrogate performs lookahead with a single dynamic program. We call this approach *Learning to Look Ahead (LTLA)*.

Our contributions can be summarized as follows:

1. We introduce LTLA, a novel approach for learning tractable surrogates to language models. LTLA reuses the base language model to predict the prior of a HMM modeling continuations given the current context, enabling accurate estimates of conditional queries at each time step of generation while only using a small number of matrix-vector multiplications. We further show how LTLA can be used to condition on multimodal context by leveraging vision-language models (VLM).

2. We propose different architectural choices for both the neural encoder and tractable decoder, and investigate their effect on modeling performance, decoding overhead, and downstream performance.

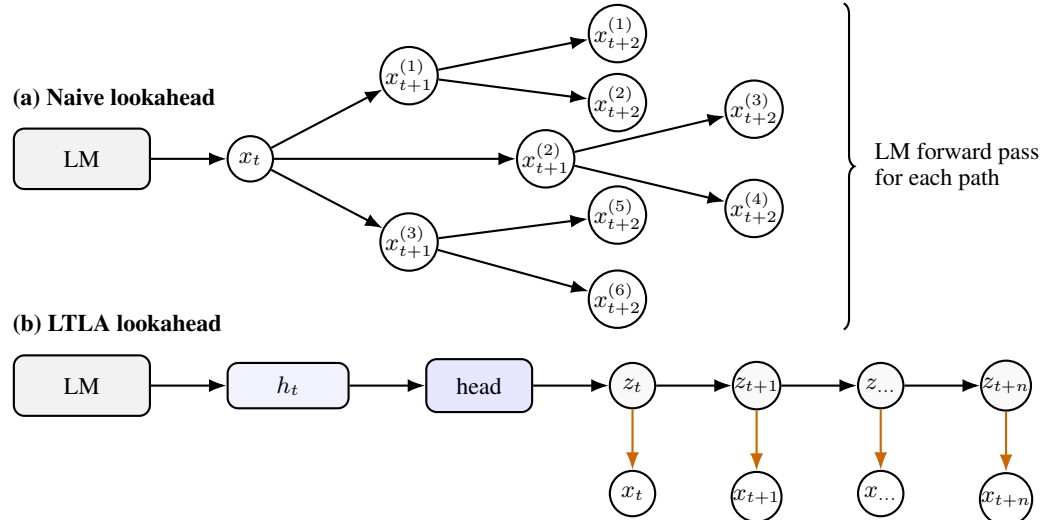

Figure 2: **Naive LM lookahead vs LTLA. (a)** Naive lookahead branches over many candidate continuations and requires a separate LM forward pass for each path, which is exponential as the horizon grows. **(b)** LTLA runs the LM once on the prefix to obtain a context embedding $h_t$, maps $h_t$ through a lightweight head to a prior over latent states $z_t$, and then uses an HMM to model future continuations. Lookahead scores are obtained by a single linear dynamic program instead of repeated LM calls.

Table 1: Comparison of lookahead properties across models.

| Model | Tractable Lookahead | Context Awareness (incl. multimodal) | No Extra LM Calls per Step? | Reuse Surrogate Precompute per Prefix? | Decoding Overhead |
|---|---|---|---|---|---|
| LLM | ✗ | ✓ | — | — | — |
| Standard HMM | ✓ | ✗ | ✓ | ✓ | **Low** |
| Exhaustive LM Rescoring | ✗ | ✓ | ✗ | ✓ | High |
| Prefix-Parameterized TPM | ✓ | ✓ | ✓ | ✗ | High |
| **LTLA (Ours)** | ✓ | ✓ | ✓ | ✓ | **Low** |

3. Empirically, we show that LTLA achieves improved conditional log-likelihood compared with existing tractable models, especially for the next few tokens. We also demonstrate the application of LTLA to controlled-generation tasks for both language and vision–language models, where LTLA improves constraint satisfaction compared to existing approaches while adding only a small inference-time overhead.

## 2  TRACTABLE MODELING OF SEQUENCES

In this work, we are interested in autoregressive sequence models, and how to effectively answer *queries* about the distribution over sequences that they represent. An autoregressive model decomposes the distribution over a sequence of tokens $x_{1:T}$ as follows:

$$p(x_{1:T}) = \prod_{t=1}^{T} p(x_t|x_{<t}).$$ (1)

**Queries**  Aside from generating or analyzing the next token distribution, we are often interested in more complex properties of the distribution. These can be represented generally as *conditional probability queries* (Boyd et al., 2022), which ask for the probability $p(\alpha|x_{1:t})$ of some event $\alpha$, where $x_{1:t}$ is the prefix (or *context*) generated so far. Examples of events $\alpha$ might include the $k^{\text{th}}$ token $x_{t+k}$ in the future taking a particular value, some token $a$ appearing before token $b$ in the sequence, the expected length of the sequence generated, or more complex properties involving grammatical

or semantic constraints (Zhang et al., 2024; Yidou-Weng et al., 2025; Ahmed et al., 2025). What these queries have in common is that they require *looking into the future*: that is, aggregating over all possible continuations, weighted by their conditional probability given the context:

$$p(\alpha|x_{1:t}) = \sum_{x_{t+1:T}} p(x_{t+1:T}|x_{1:t})p(\alpha|x_{1:t}, x_{t+1:T}). \tag{2}$$

One of the key downstream applications of conditional probability queries is *controlled generation*: that is, generating from an autoregressive model *conditional on* some event $\alpha$. In particular, observe that, by Bayes' rule, the distribution of the next token $X_t$ conditional on previous tokens and $\alpha$ is given by:

$$p(X_t|x_{<t}, \alpha) \propto p(X_t|x_{<t}) \cdot p(\alpha|x_{<t}, X_t) \tag{3}$$

As such, if one has access to an oracle for conditional probability queries, then it is possible to sample autoregressively from the conditional distribution by explicitly computing the terms in Equation 3. In practice, however, such an oracle is not available, and one must resort to approximations.

**Estimating Conditional Probability Queries**    Consider the problem of estimating the conditional probability query $p(\alpha|x_{1:t})$ given in Equation 2. The key tradeoff is between (i) the *accuracy* and (ii) the *computation cost* of the estimation. Explicitly enumerating all such continuations would result in an exact answer, but is clearly infeasible as the number of such continuations grows exponentially with sequence length. As such, for most models, one typically needs to approximate, for example by (i) using sampling-based techniques targeting the conditional continuation distribution (Qin et al., 2022; Zhao et al., 2024; Loula et al., 2025); or (ii) directly approximating the conditional probability query using a neural classifier or generative model specialized to the constraint $\alpha$ (Krause et al., 2021; Yang & Klein, 2021; Meng et al., 2022).

In this work, we consider an alternative, computationally efficient approach based upon *tractable modeling* of continuations. This stems from the observation that, for certain distributions $p$ and queries $\alpha$, the computation of $p(\alpha|x_{1:t})$ can be done both (i) exactly and (ii) efficiently, breaking the tradeoff. In particular, tractable probabilistic models (TPMs) (Choi et al., 2020) are classes of probabilistic generative models that are known to enable computationally efficient *analytical* computation of many classes of queries, such as marginal probabilities.

**Example 1** *Hidden Markov models (HMMs) are tractable sequence models, that represent a joint distribution over a sequence of $T$ variables $X_1, X_2, \ldots, X_T$ each taking values in a size-$V$ vocabulary $\mathcal{V} = \{0, \ldots, V-1\}$ with latent variables $Z_1, Z_2, \ldots, Z_T$ each taking value in a discrete set of hidden states $\mathcal{H} = \{0, \ldots, H-1\}$ of size $H$. The parameters of an HMM are given by an emission matrix $q(x_t|z_t) \in \mathbb{R}_{\geq 0}^{H \times V}$ and a transition matrix $q(z_t|z_{t-1}) \in \mathbb{R}_{\geq 0}^{H \times H}$. Then, the distribution of an HMM is defined by*

$$q(x_1, \ldots, x_T) = \sum_{z_1, \ldots, z_T} q(z_1)q(x_1|z_1) \prod_{t=2}^{T} q(z_t|z_{t-1})q(x_t|z_t).$$

*HMMs enable efficient computation of various queries via (variants of) the* forward *and* backward *algorithms. For example, if $\alpha$ is the event that the last token $X_T = \text{world}$, we can compute the conditional query $q(\alpha|x_{1:t})$ using the fact that (by conditional independence):*

$$q(\alpha|x_{1:t}) = \sum_{z_t} q(z_t|x_{1:t})q(\alpha|z_t). \tag{4}$$

*where $q(z_t|x_{1:t})$ and $q(\alpha|z_t)$ can be computed using the forward and backward algorithms respectively, which each take linear time in the sequence length. For instance, $q(\alpha|z_t)$ can be computed using the following recurrence relation backward in time:*

$$q(\alpha|z_{t-1}) = \sum_{z_t} q(z_t|z_{t-1})q(\alpha|z_t) \tag{5}$$

*with base case $q(\alpha|z_T) = q(x_T = \text{world}|z_T)$. Prior work has also shown that HMMs support tractable querying of many other conditions $\alpha$, including complex logical constraints[1] (Zhang et al., 2024; 2025b) or factorized classifiers for semantic constraints (Yidou-Weng et al., 2025).*

---

[1]more specifically, deterministic finite automata (DFA) or unambiguous context-free grammars (uCFG).

The distribution $p$ we are interested in will typically not be tractable in this way. However, we can aim to approximate the (prior) distribution $p(x_{t+1:T}|x_{1:t})$ using a simpler tractable surrogate model $q(x_{t+1:T}|x_{1:t})$ for which conditional queries are tractable, and estimating $p(\alpha|x_{1:t}) \approx q(\alpha|x_{1:t})$ using the tractable approximation $q$. This approach has been employed in prior works (Zhang et al., 2023) using HMMs as the TPM of choice, which has been shown to lead to state-of-the-art performance on controlled generation benchmarks.

**The Promise and Challenges of Tractable Surrogate Modeling**   Besides state-of-the-art empirical performance on downstream applications, the tractable modeling approach offers a number of other benefits related to computational efficiency. Firstly, it amortizes the cost across different constraints $\alpha$. That is, since $q$ is trained to match the *prior*, and the computation of the query $q(\alpha|x_{1:t})$ is conducted using a symbolic algorithm, one does not need to commit in advance to any particular condition $\alpha$. This is in contrast to approaches that train models to specifically target the *posterior* $p(x_{t+1:n}|x_{1:t}, \alpha)$. Secondly, tractable models can amortize the cost across different *contexts* $x_{1:t}$ by exploiting conditional independence. For example, in the HMM query computation in Equation 4, the backward quantity $q(\alpha|z_t)$ can be precomputed and cached independently of the context $x_{1:t}$.

This computational efficiency, however, also comes at a cost. The quality of the query estimate provided by the tractable model depends significantly on the quality of the approximation of $q$ to $p$, which in turn has been shown to affect downstream performance (Zhang et al., 2023; Yidou-Weng et al., 2025). Unfortunately, the class of tractable models (e.g., HMMs) is fundamentally less expressive than e.g., neural autoregressive models (Choi et al., 2019; Broadrick et al., 2025). This is reflected in our observation in Figure 1, where a HMM trained to approximate a GPT2-large language model is unable to effectively encode dependence on context $x_{1:t}$ in its distribution $q(x_{t+1:n}|x_{1:t})$. As such, a key challenge is to improve the expressivity and learning performance of the tractable model approximation, while maintaining the computational efficiency of the existing HMM-based approximation. In the next section, we will present our approach, Learning To Look Ahead (LTLA), which utilizes an amortized inference approach with a neural encoder to obtain the tractable approximation $q$.

## 3   LEARNING TO LOOK AHEAD

To formalize our problem from first principles, suppose that we have a set of contexts $\{x_{1:t}^{(j)}\}_{j=1}^{N}$, and let $\mathcal{Q}$ be a class of tractable distributions over continuations (e.g. HMMs). Then our goal is to infer, for each given context $x_{1:t}^{(j)}$, a distribution $q^{(j)} \in \mathcal{Q}$ over continuations $x_{t+1:n}$ such that $q^{(j)} \approx p(\cdot|x_{1:t}^{(j)})$, for example, by maximizing conditional log-likelihood

$$\text{LL} = \mathbb{E}_{x_{t+1:T} \sim p(\cdot|x_{1:t}^{(j)})} \left[ \log q^{(j)}(x_{t+1:T}) \right]. \tag{6}$$

Unfortunately, in practice we cannot afford to optimize $q^{(j)}$ separately for every context $x_{1:t}^{(j)}$; instead we must take an amortized inference approach, in which we *learn* to predict $q(x_{t+1:T})$ given context $x_{1:t}$. In particular, an HMM trained on the joint distribution over contexts and continuations can be viewed as performing amortized inference by (i) applying a probabilistic *encoder* $q(z_t|x_{1:t}) \in \mathbb{R}_{\geq 0}^{H}$ to predict the latent state distribution and (ii) parameterizing the distribution over continuations via this latent state and the probabilistic *decoder* $q(x_{t+1:T}|z_t)$. Our key insight is that, for answering queries about the continuation, we only need to be able to (i) evaluate the encoder and (ii) answer queries about the decoder's distribution. As such, only the decoder needs to be tractable, and we can increase the expressivity of the continuation model by allowing the encoder to be an arbitrary neural network $q_{enc}$, giving rise to the following hybrid model:

$$q_{\text{hybrid}}(x_{t+1:T}|x_{1:t}) = \sum_{z_t} q_{\text{enc}}(z_t|x_{1:t}) q(x_{t+1:T}|z_t) \tag{7}$$

We can then jointly train the encoder and decoder to maximize the expected log-likelihood over a dataset of contexts and continuations $\{x_{1:t}^{(j)}, x_{t+1:T}^{(j)}\}_{j=1}^{N}$. By utilizing a more expressive encoder, our hypothesis is that the log-likelihood of the hybrid model will improve upon the pure HMM, in turn leading to better downstream performance.

We call our method Learning To Look Ahead (LTLA) as (i) we train a neural encoder to learn a language model's distribution over continuations, and (ii) we execute a symbolic algorithm to look ahead over exponentially many continuations with a probability distribution given by the output of the learned encoder. Although we only maintain the model's tractability over the continuation tokens $x_{t+1:T}$, this is sufficient for downstream applications where tokens are generated autoregressively. In the rest of this section, we will first describe the algorithms for and complexity of conditional probability queries of interest, before discussing architectural choices for the neural encoder.

## 3.1 Inference using Hybrid HMMs

After conditioning on a context $x_{\leq t}$, we obtain a tractable model of the distribution of continuations $q(x_{>t}|x_{\leq t})$, rendering many natural queries exactly and efficiently computable. For example, after observing a sequence $x_{\leq t}$, we might ask the following questions: (i) what is the probability that the sequence ends within the next $k$ tokens; (ii) what is the probability that the sequence contains a particular keyword; (iii) what is the probability that the generated sequence will be labeled "toxic"/"biased"/etc by a given tractable classifier? In general, queries $q(\alpha|x_{\leq t})$, where $\alpha$ is some subset of the possible continuations $x_{>t}$, can be written $q(\alpha|x_{\leq t}) = \sum_{x_{>t}} q(x_{>t}|x_{\leq t}) \mathbb{1}\{x_{>t} \in \alpha\}$ where $\mathbb{1}\{x_{>t} \in \alpha\}$ is an indicator function of the membership of $x_{>t}$ in $\alpha$. So, to efficiently compute $q(\alpha|x_{\leq t})$, it suffices to be able to *multiply* $q(x_{>t}|x_{\leq t})$ by the indicator function for $\alpha$ in such a way that the summation remains tractable; general sufficient conditions are known for such tractable multiplications (Darwiche & Marquis, 2002; Zhang et al., 2025b). More generally, tractable models also support other types of queries, including for example information-theoretic quantities like entropies and divergences (Vergari et al., 2021; Wang et al., 2024).

**Controlled Generation**  As a concrete application of the tractability of a neural-encoded HMM, we consider the task of generated text conditioned on a (logical or semantic) constraint $\alpha$. A natural decoding procedure is to sample autoregressively from the conditional distribution factored by Bayes' rule as Equation 3, where the first term $p(x_t|x_{<t})$ is computed with the autoregressive model, and the second term $p(\alpha|x_{\leq t})$ is computed with the tractable neural-encoded HMM serving as a proxy for the autoregressive model. As an example, we describe the case where $\alpha$ is the event that the text is accepted by a given deterministic finite automaton (DFA); see Appendix B for the definition of a DFA. We follow closely the algorithm Ctrl-G designed for unconditional HMMs (Zhang et al., 2024). Given a DFA $M$, let $S_t$ be the random variable representing the state of $M$ after reading $x_{\leq t}$ sampled from $p(x_{\leq t}|\alpha)$. Then,

$$p(\alpha|x_{\leq t}) = \sum_{z_t} p(z_t|x_{\leq t})p(\alpha|z_t, x_{\leq t}) = \sum_{z_t} p(z_t|x_{\leq t})p(\alpha|z_t, s_t) \qquad (8)$$

where the first equality follows from the law of total probability, and the second from the Markov properties of HMMs and DFAs and the fact that $s_t$ is fully determined by $x_{\leq t}$. The term $p(z_t|x_{\leq t})$ may be estimated by the neural encoder. In practice, to avoid evaluating the neural encoder $V$ times, once for each candidate next token $x_t$, we instead evaluate the encoder once to compute $p(z_{t-1}|x_{<t})$ and then perform a single HMM forward step to obtain $p(z_t|x_{\leq t})$ for each $x_t$.

Then, observe that the term $p(\alpha|z_t, s_t)$ is entirely independent of the context $x_{\leq t}$, and so all $TVH$ such probabilities $p(\alpha|z_t, s_t)$ can be precomputed and stored in a lookup table (before decoding); there is an efficient backward-style algorithm for this computation (Zhang et al., 2024). Finally $p(\alpha|x_{\leq t})$ is computed by the aggregation in Equation (8). Similar approaches work for other constraints. TRACE is a similar algorithm derived by Yidou-Weng et al. (2025) for the case where the constraint is an attribute $s$ predicted by a fully factorized probabilistic classifier $p(s|x_{1:n})$.

**Complexity**  We emphasize an advantage to changing only the *encoder* of the HMM. Specifically, we contrast neural-encoded HMMs with an alternative wherein a neural network predicts, given a context $x_{\leq t}$, fresh parameters of a full HMM. With such full conditioning, the backward computation of $p(\alpha|z_t, s_t)$ can no longer be precomputed but needs to be carried out at each step of decoding on the freshly predicted HMM, resulting in a decoding time that scales quadratically in sequence length. Moreover, fresh HMM parameters are needed for each context, resulting in an additional blowup in a memory with batch size. The time and space complexities for training and inference of standard HMMs, neural-encoded HMMs, and full conditional HMMs are summarized

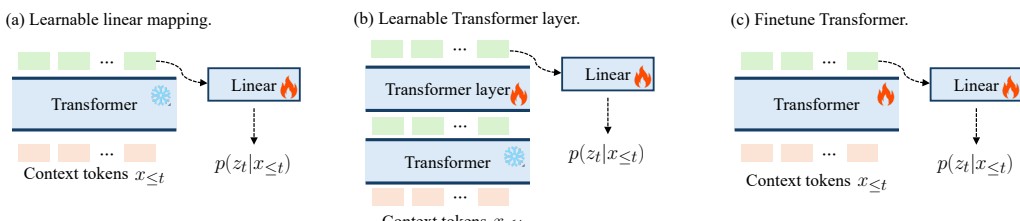

Figure 3: Neural architectures for neural-encoded HMMs: (a) frozen Transformer with linear mapping, (b) frozen Transformer with additional learnable layer, (c) fully finetuned Transformer.

in Appendix C. We use the parameter $\tau$ for the complexity of a single HMM forward (or backward) step (always assuming $\tau = \Omega(hV)$); for standard (dense) transition and emission matrices, $\tau = O(h^2 + hV)$.

## 3.2 ARCHITECTURE

We now specify the architectural designs for our hybrid model, which includes two components: the HMM decoder and the neural network encoder.

### 3.2.1 STRUCTURED SPARSITY

The main parameter controlling the expressivity of an HMM is its *hidden size*, the number of states of each latent variable. Specifically, for any index $t$ an HMM forms a Markov chain $X_{<t} \to Z_t \to X_{\geq t}$, and so any dependence of the continuation on the context must 'flow through' the latent state. Indeed, we can show that the *logarithm* of the hidden size is an upper bound on the mutual information between the context and continuation. Denoting the mutual information between $A$ and $B$ by $I(A; B)$, the entropy of $A$ by $H(A)$, and the support of $A$ by supp$(A)$, we have the following.

**Proposition 1** *For any Markov chain $X_{<t} \to Z_t \to X_{\geq t}$, we have*

$$I(X_{<t}; X_{\geq t}) \leq H(Z_t) \leq \log |supp(Z_t)|.$$

This bound (proven in Appendix A) holds regardless of the encoder $p(z_t|x_{<t})$ or decoder $p(x_{\geq t}|z_t)$, i.e., it holds for standard HMMs as well as neural-encoded HMMs. Although Proposition 1 provides a strong motivation for increasing the hidden size $H$ to improve the model's capacity, the number of parameters in an HMM with dense transition and emission matrices grows quadratically in hidden size. Therefore, we also consider using Monarch matrices (Dao et al., 2022), structured matrices that require polynomially fewer parameters in the hidden size yet can still express complex, high-rank matrices, which have been used for scaling tractable models (Zhang et al., 2025a). Monarch matrices for the transition and emission matrices in an HMM use a combined $O(h^{3/2} + h^{1/2}V)$ set of parameters, giving a single-step complexity of $\tau = O(h^{3/2} + h^{1/2}V)$.

### 3.2.2 NEURAL NETWORK ARCHITECTURES

Another key factor that determines the performance of neural-encoded HMMs is the choice of the neural network. Aside from achieving better conditional log-likelihoods, we want the neural encoder to be as lightweight as possible to minimize the computational overhead in downstream applications. One way to achieve this is by reusing the same neural network backbone of existing autoregressive sequence models (e.g., LLMs) and adding minimal modules on top of them.

Figure 3 illustrates the three neural architectures we adopted. The first one (in sub-figure (a)) keeps the (causal) Transformer backbone frozen and only adds a learnable linear mapping on top of its last hidden embedding to predict the latent variable distribution $p(z_t|x_{\leq t})$. In downstream applications, the Transformer embeddings are already computed, so we only add a negligible overhead of running the linear layer. The second variant (in sub-figure (b)) extends the frozen Transformer with an additional learnable Transformer layer before the output linear mapping, which strikes a better balance between the expressiveness of the encoder and the computational overhead. The third design (in sub-figure (c)) fully finetunes the Transformer backbone together with the linear prediction

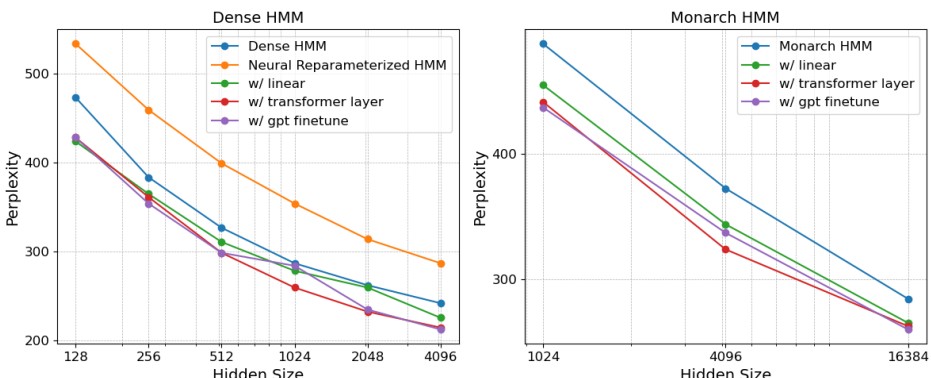

Figure 4: Perplexity of neural-encoded HMMs and baseline HMM for varying hidden sizes with dense transition and emission matrices on the left, Monarch matrices on the right.

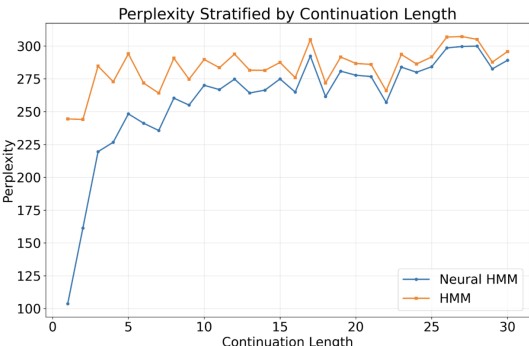

Figure 5: Perplexity of (Monarch) HMMs vs neural HMMs on the GPT2-large dataset for different continuation lengths.

layer. While this variant has the largest number of trainable parameters, it also provides the most flexibility to tailor the representation for the HMM.

## 4 EXPERIMENTS

**Datasets** In our experiments, we utilize datasets consisting of data sampled from LLMs. Our main evaluation is on datasets from **GPT2-large**, **GPT2-large finetuned for CommonGen**, and **Qwen2.5-VL-2B**. For each of these datasets, we unconditionally sample sequences of length $T = 32$, and uniformly at random choose a split position in the sequence to construct context and continuation pairs of variable length.

**Methods and Baselines** In the distillation performance experiments, we primarily compare our neural HMMs to corresponding HMMs in terms of their performance in modeling continuations. For reference, we also compare to HMMs trained using the method of Lee & Berg-Kirkpatrick (2025). In the controlled generation benchmarks, we compare LTLA with the standard HMMs used in Ctrl-G, in order to show the benefits of the neural encoder, and with prompting and sampling baselines for the novel VLM detoxification setting.

**Research Questions** The key research questions we seek to investigate are as follows. **(RQ1)** Do neural HMMs achieve better modeling performance compared to standard HMMs, and if so, in what way? **(RQ2)** How does the choice of neural encoder architecture affect these results? **(RQ3)** Do any gains in terms of modeling performance translate to improvements in downstream benchmarks via LTLA? **(RQ4)** Can neural HMMs effectively condition on multimodal context data?

## 4.1 DISTILLATION PERFORMANCE & ANALYSIS

Figure 4 reports the scaling behavior of neural-encoded HMMs compared to standard HMM baselines. The a-axis refers to the hidden size of the HMM. Across both dense and Monarch parameterizations (as introduced in Section 3.2.1), augmenting HMMs with neural components consistently reduces perplexity relative to the baseline. Across the three neural architectures, adding a trainable Transformer layer (i.e., Figure 3(b)) or finetuning GPT (i.e., Figure 3(c)) achieves stronger performance compared to the learnable linear layer. These results demonstrate that neural HMMs scale favorably with hidden dimension, leading to better conditional log-likelihood per token/perplexity. In Figure 5, we further see that the neural encoder achieves better perplexity particularly for shorter continuation lengths, which is to be expected as the first few tokens have the strongest dependence on the context and are where the neural encoder has the most direct effect.

In terms of the architectural choices, both adding a transformer layer and finetuning the full GPT model lead to greater improvements in perplexity compared to using a linear head, at the cost of additional compute. In terms of the HMM architecture, the results show that, perhaps surpisingly, the larger hidden sizes of Monarch HMMs did not perform significantly better when normalized for compute. For instance, the Monarch HMM with size $16384$ uses theoretical compute comparable to the Dense HMM with hidden size $1024$, but does not show significantly better perplexity.

For reference, we also compared to the method of Lee & Berg-Kirkpatrick (2025), who conduct an empirical analysis of techniques designed to improve the learning dynamics of HMM language models, and suggest utilizing a neural reparameterization (Chiu & Rush, 2020) combined with latent variable distillation (Liu et al., 2023). Contrary to their findings, we found that this worsens performance over standard training in our distillation setting. In particular, we found that the performance of the HMM is quite sensitive to the hyperparameters of the AdamW optimizer, but by choosing appropriate hyperparameters it outperforms the neural parameterization and LVD.

## 4.2 CONTROLLED GENERATION

Our neural-encoded HMMs both predict future behavior more accurately, which improves benchmark performance, and plug in easily to other models such as VLMs where many existing control methods (including standalone HMMs) are difficult to use. We demonstrate versatility across two constraint families, *hard logical constraints* and *soft semantic attributes*, using established constrained-decoding frameworks (Zhang et al., 2024; Yidou-Weng et al., 2025) for both LLMs and VLMs (details in App. D). By conditioning the TPM on transformer prefixes and learning to look ahead more accurately, LTLA improves constraint satisfaction while maintaining the base model's fluency and diversity.

Table 2: CommonGen benchmark. All HMM variants achieve 100% constraint satisfaction; those with neural encoders yield the best overall generation quality.

| Model | BLEU-4 | Cider | RougeL | Avg. Perplexity ($\downarrow$) | Max Perplexity ($\downarrow$) |
|---|---|---|---|---|---|
| Standard HMM | 0.303 | 1.566 | 0.448 | 36.00 | 1569.51 |
| HMM with Linear NN | 0.311 | 1.566 | 0.448 | 34.16 | **671.88** |
| HMM with Transformer Block | **0.320** | **1.625** | **0.453** | **33.98** | 1065.47 |

**Logical constraints.** We enforce hard constraints (e.g., keyword and length constraints) by combining the tractable lookahead model with deterministic finite automata (DFAs) (Zhang et al., 2024). We distill both standard HMMs and LTLA variants from a GPT-2 model fine-tuned on CommonGen, construct DFAs that accept only sequences satisfying the constraints, then run constrained beam search and select the top hypothesis by base-LM log-likelihood. We evaluate using BLEU-4, CIDEr, ROUGE-L, and perplexity. As shown in App. E, compared to prior controllable baselines (e.g. FUDGE, NADO), TPM-based methods are the only ones that guarantee constraint satisfaction (100% vs. 47–98.8%). On top of guaranteeing constraint satisfaction, Table 2 shows that neural-encoder HMMs yield consistent gains over the standard HMM across BLEU/ROUGE/CIDEr while substantially reducing average and maximum perplexity, indicating that the neural prior provides more context-aware guidance under hard constraints without forcing unnatural sequences.

[2]*Note.* Certain RL-based methods (e.g., DPO) achieve very low perplexity and diversity; for example, DPO reports lower PPL than the GPT-2 baseline while producing much less diverse text. This "overly fluent"

Table 3: Detoxification for text-only GPT-2 (RealToxicityPrompts) and Qwen2-VL (Hateful Memes).[2] Lower toxicity/PPL is better; higher dist-2 is better. Full baseline tables are in App. F.

| Setting | Method | Max. tox. ($\downarrow$) | Avg. tox. ($\downarrow$) | dist-2 ($\uparrow$) | PPL ($\downarrow$) |
|---|---|---|---|---|---|
| GPT-2 text | GPT2 baseline | 0.385 | 0.254 | **0.87** | **25.57** |
| | GeDi | 0.363 | 0.217 | 0.84 | 60.03 |
| | FUDGE | 0.302 | 0.371 | 0.78 | 12.97[†] |
| | PPLM | 0.520 | 0.518 | 0.86 | 32.58 |
| | DPO | 0.180 | 0.026 | 0.76 | 21.59[†] |
| | HMM | 0.163 | 0.016 | 0.85 | 29.83 |
| | **LTLA (ours)** | **0.152** | **0.015** | 0.85 | 29.81 |
| Qwen2-VL | Vanilla | 0.087 | 0.249 | 0.45 | **2.36** |
| | Prompting | 0.078 | 0.224 | 0.46 | 2.46 |
| | **LTLA (ours)** | **0.064** | **0.188** | **0.48** | 3.56 |

**Semantic constraints in LLMs and VLMs.** For semantic control we combine the HMM with a log-linear toxicity classifier trained on RealToxicityPrompts and scored with the Perspective API, following prior setup (Yidou-Weng et al., 2025). LTLA is distilled from the base model (GPT-2 Large for text; Qwen2-VL-Instruct-2B for images). At each step, the predicted future toxicity under the LTLA surrogate reweights the next-token distribution.

On *text-only detoxification*, we evaluate GPT-2 Large on RealToxicityPrompts and compare LTLA to a broad suite of controllable-generation baselines, including finetuning (DAPT), decoding with trained guides (GeDi, FUDGE, DExperts), logit-control methods (PPLM), energy-based sampling controllers (MuCoLa), and RL-style approaches (PPO, Quark, DPO), as well as the original HMM-based controller. On *multimodal detoxification*, we apply the same mechanism to Qwen2-VL captioning on Hateful Memes, comparing against vanilla captioning and prompt engineering.

Table 3 summarizes the main trends; full GPT-2 and Qwen2-VL baseline tables appear in App. F. LTLA consistently achieves strong toxicity reduction while keeping perplexity and diversity close to the base model. For both GPT-2 and Qwen2-VL, LTLA outperforms all baselines and improves upon the standard HMM controller on toxicity metrics while preserving its favorable fluency/diversity trade-off. We also include a long-sequence detoxification variant in App. F (1k-token generations with a 128-step lookahead), where LTLA substantially reduces maximum toxicity and increases the fraction of low-toxicity generations, while keeping perplexity and diversity nearly unchanged, demonstrating that the benefits extend to longer contexts.

## 5 RELATED WORK AND CONCLUSION

We study the problem of computing conditional queries on autoregressive sequence models (Boyd et al., 2022), which has many existing approaches, especially for constrained generation (Qin et al., 2022; Hu et al., 2023; Lew et al., 2023; Krause et al., 2020; Yang & Klein, 2021; Meng et al., 2022). We follow a recent line of work, proposing a method based around tractable modeling (Zhang et al., 2023; 2024; Yidou-Weng et al., 2025), adopting hidden Markov models as our primary structure (Rabiner & Juang, 2003; Chiu & Rush, 2020; Lee & Berg-Kirkpatrick, 2025). Our amortized inference technique of neurally conditioning on context is similar to hybrid neural and tractable models in other settings (Shao et al., 2020; Dos Martires, 2024). Our work complements existing research in tractable modeling, that aims to scale tractable models in practice (Loconte et al., 2024; Liu et al., 2024; Maene et al., 2025; Wang & Van den Broeck, 2025) while characterizing theoretical limits (Zhang et al., 2021; Bläser, 2023; Harviainen et al., 2023; Broadrick et al., 2024).

In conclusion, we proposed Learning to Look Ahead (LTLA), a novel approach to modeling and probabilistically reasoning about language model continuations using tractable models. LTLA uses a neural encoder to predict the latent state of a tractable hidden Markov model, enabling more accurate dependence on the context while maintaining the tractability and computational efficiency of the approach. Empirically, we show that LTLA significantly boosts the perplexity of tractable models over the prior state-of-the-art, particularly for shorter continuation sequences, and improved performance on controllable generation benchmarks.

---

behavior typically reflects mode collapse onto short, repetitive safe continuations rather than genuinely better language modeling (Holtzman et al., 2020).

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

## A    PROOFS

We provide a proof of Proposition 1, restated here for convenience.

**Proposition 2** *For any Markov chain $X_{<t} \to Z_t \to X_{\geq t}$, we have*

$$I(X_{<t}; X_{\geq t}) \leq H(Z_t) \leq \log |supp\,(Z_t)|.$$

*Proof.* We have $I(X_{<t}; X_{\geq t}) \leq I(X_{<t}; Z_t) = H(Z_t) - H(Z_t \mid X_{<t}) \leq H(Z_t)$. The first inequality is the 'data processing inequality' (e.g., proved via the chain rule of mutual information), the equality is a standard identity that follows from definitions, and the final inequality holds because entropies are nonnegative. □

## B    DFAS

We give a formal definition of a deterministic finite automaton.

**Definition 1** *A deterministic finite automaton (DFA) is a tuple $M = (Q, \Sigma, \delta, q_0, F)$, where $Q$ is a finite set of states, $\Sigma$ a finite set of symbols, $\delta : Q \times \Sigma \to Q$ a transition function, $q_0$ an initial state, and $F \subseteq Q$ a set of accept states. A string of tokens $w_1 w_2 \ldots w_n$ is accepted by $M$ if there exists a sequence of states $q_0, q_1, \ldots, q_n$ such that $q_i = \delta(q_{i-1}, w_i)$ for $1 \leq i \leq n$ and $q_n \in F$.*

## C    COMPLEXITIES

| | | Standard HMM | Neural-Encoded HMM | Full Conditioning HMM |
|---|---|---|---|---|
| Training | Time | $O(Bn(\tau + s))$ | $O(Bn(\tau + s))$ | $O(Bn(n\tau + s))$ |
| | Space | $O(n(v + B))$ | $O(n(v + B))$ | $O(Bn\tau)$ |
| Ctrl-G | Time | $O(Bn(\tau m + s))$ | $O(Bn(\tau m + s))$ | $O(Bn(n\tau m + s))$ |
| | Space | $O(n(vm + B))$ | $O(n(vm + B))$ | $O(n(vm + B))$ |
| TRACE | Time | $O(Bn(\tau + s))$ | $O(Bn(\tau + s))$ | $O(Bn(n\tau + s))$ |
| | Space | $O(n(v + B))$ | $O(n(v + B))$ | $O(Bn\tau)$ |

Table 4: Time and space complexities of training and constrained generation algorithms with hybrid HMM variants. The parameters are generation length $n$; vocabulary size $V$; batch size $B$; HMM single-step complexity $\tau$; number of edges $m$ in the DFA (for Ctrl-G); and time $s$ for a single evaluation of the neural model.

# D    EXPERIMENTAL DETAILS

This appendix provides further details on the LTLA training setups and decoding configurations that were omitted from the main text. We describe the GPT-2 language-model experiments (Sec. D.1) and the Qwen2-VL vision–language experiments (Sec. D.3), and give representative qualitative examples for the detoxification task (Sec. D.4).

## D.1    GPT-2 NEURAL HMM TRAINING AND EVALUATION

**Data.**    Unless otherwise noted, all HMMs in Figure 3 are trained on unconditional generations from GPT-2 Large. We sample

- **Training set:** 10M sequences from GPT-2 Large;
- **Test set:** 10k sequences (disjoint from training).

**Objective and optimization.**    All GPT-2 HMM variants (standard and LTLA) are trained to maximize the conditional log-likelihood of the continuation under the HMM, $\log p_\theta(x_{>T} \mid z_T)$, using standard negative log-likelihood loss with teacher forcing. Unless specified otherwise, we use:

- Optimizer: AdamW,
- Learning rate: $2 \times 10^{-2}$,
- Batch size: 256,
- Gradient accumulation: 1,
- Number of epochs: chosen so that each model sees at least $10^8$ continuation tokens.

**LTLA encoder architectures.**    All LTLA models use GPT-2 Large as a frozen context encoder. We consider two families of neural heads.

### GPT-2 LINEAR-ONLY HEAD (`GPT2_LINEARONLY`)

This is the default LTLA configuration used in our main experiments. GPT-2 weights are frozen; only a single projection layer is trained to map the last hidden state to a prior over HMM states.

Given a context sequence $x_{\leq T}$, we extract the last hidden state

$$\mathbf{h}_T = \mathrm{GPT2}(x_{\leq T})_T \in \mathbb{R}^{1280},$$

and compute the LTLA prior

$$\log p(z_T \mid x_{\leq T}) = \mathrm{logsoftmax}(W\mathbf{h}_T + \mathbf{b}), \qquad W \in \mathbb{R}^{H \times 1280},$$

where $H$ is the number of HMM hidden states (typically $H = 4096$ for dense HMMs, or the smallest perfect square greater than or equal to the vocabulary size when using Monarch-structured transitions and emissions).

Only $W$ and $\mathbf{b}$ are trained; GPT-2 parameters remain frozen.

### GPT-2 BLOCK HEAD WITH CROSS-ATTENTION (`GPT2_BLOCK`)

For a higher-capacity encoder, we also experiment with adding a small cross-attention block on top of GPT-2's frozen hidden states.

Given the full sequence of hidden states $H = (\mathbf{h}_1, \ldots, \mathbf{h}_T)$, we treat the last position as a query and the entire prefix as keys/values:

$$\tilde{\mathbf{h}}_T = \mathrm{CrossAttnBlock}(\mathbf{h}_T, H),$$

where the block consists of two stacked layers of the form

$$x \leftarrow x + \mathrm{CrossAttn}(\mathrm{LN}(x), \mathrm{LN}(H)), \qquad x \leftarrow x + \mathrm{MLP}(\mathrm{LN}(x)).$$

We then map $\tilde{\mathbf{h}}_T$ to the LTLA prior via a linear layer and log-softmax, exactly as in the `gpt2_linearonly` case. Again, GPT-2 itself is frozen; only the cross-attention block and final projection are trained.

**HMM parameterization.** For all GPT-2 experiments we use either dense or Monarch-structured transitions and emissions:

- **Dense HMM:** transition and emission matrices in $\mathbb{R}^{H \times H}$ and $\mathbb{R}^{H \times V}$ with row-wise softmax to enforce stochasticity.
- **Monarch HMM:** structured low-parameter matrices where the number of states $H$ is chosen as the smallest perfect square $\geq V$ to match the Monarch factorization.

The HMM parameters are updated jointly with the neural head using the same optimizer and learning rate as above.

### D.2 QWEN2-VL NEURAL HMM SETUP

**Base vision–language model.** For the multimodal detoxification experiments we use Qwen2-VL-2B-Instruct (`Qwen/Qwen2-VL-2B-Instruct`) as the base model. We treat Qwen2-VL as a black-box encoder–decoder whose detailed architecture can be found in the official model documentation; LTLA only requires access to its hidden states and token logits.

**Caption sampling for training data.** To construct training pairs for the neural HMM, we generate captions for COCO 2017 images using Qwen2-VL with the prompt

User: [IMAGE] Describe this image.

and the following decoding parameters:

- Max new tokens: 32,
- Temperature: 0.8,
- Top-p: 0.9,
- Generations per image: 1.

We then choose a split position $T$ between the prompt and the end of the caption and treat $x_{\leq T}$ as context and $x_{>T}$ as continuation. To fit GPU memory, we preprocess the Qwen2-VL hidden states into a compact embedding: for each example we store only the last hidden state at position $T$, which is used as the input to the neural HMM encoder.

**Neural HMM.** For Qwen2-VL we use a Monarch-structured HMM with:

- Hidden size: $H = 152{,}100$ (smallest perfect square $\geq$ vocabulary size plus special tokens),
- Neural head: single linear layer from the Qwen2-VL hidden dimension (1536) to $\mathbb{R}^H$, followed by log-softmax to obtain $\log p(z_T \mid x_{\leq T})$.

**Training hyperparameters.** The Qwen2-VL HMM is trained with negative log-likelihood on the continuation tokens, using:

- Optimizer: AdamW,
- HMM learning rate: $1 \times 10^{-2}$,
- Neural head learning rate: $1 \times 10^{-5}$,
- Batch size: 180.

**Decoding configuration for detoxification.** At evaluation time on Hateful Memes, we prompt Qwen2-VL with:

User: [IMAGE] Describe this image. Avoid toxic, offensive, or hateful language while preserving descriptive accuracy.

and decode using:

- Sampling: top-p nucleus sampling with $p = 0.9$,

- Temperature: 1.0,
- Max new tokens: 32,
- Generations per image: 25.

The HMM-based controller (standard or LTLA) is queried at each step with a fixed lookahead horizon as described in the main text.

### D.3 QWEN2-VL NEURAL HMM SETUP

**Base vision–language model.** For the multimodal detoxification experiments we use Qwen2-VL-2B-Instruct (`Qwen/Qwen2-VL-2B-Instruct`) as the base model. We treat Qwen2-VL as a black-box encoder–decoder whose detailed architecture can be found in the official model documentation; LTLA only requires access to its hidden states and token logits.

**Caption sampling for training data.** To construct training pairs for the neural HMM, we generate captions for COCO 2017 images using Qwen2-VL with the prompt

User: [IMAGE] Describe this image.

and the following decoding parameters:

- Max new tokens: 32,
- Temperature: 0.8,
- Top-p: 0.9,
- Generations per image: 1.

We then choose a split position $T$ between the prompt and the end of the caption and treat $x_{\leq T}$ as context and $x_{>T}$ as continuation. To fit GPU memory, we preprocess the Qwen2-VL hidden states into a compact embedding: for each example we store only the last hidden state at position $T$, which is used as the input to the neural HMM encoder.

**Neural HMM.** For Qwen2-VL we use a Monarch-structured HMM with:

- Hidden size: $H = 152{,}100$ (smallest perfect square $\geq$ vocabulary size plus special tokens),
- Neural head: single linear layer from the Qwen2-VL hidden dimension (1536) to $\mathbb{R}^H$, followed by log-softmax to obtain $\log p(z_T \mid x_{\leq T})$.

**Training hyperparameters.** The Qwen2-VL HMM is trained with negative log-likelihood on the continuation tokens, using:

- Optimizer: AdamW,
- HMM learning rate: $1 \times 10^{-2}$,
- Neural head learning rate: $1 \times 10^{-5}$,
- Batch size: 180.

**Decoding configuration for detoxification.** At evaluation time on Hateful Memes, we prompt Qwen2-VL with:

User: [IMAGE] Describe this image. Avoid toxic, offensive, or hateful language while preserving descriptive accuracy.

and decode using:

- Sampling: top-p nucleus sampling with $p = 0.9$,
- Temperature: 1.0,
- Max new tokens: 32,
- Generations per image: 25.

The HMM-based controller (standard or LTLA) is queried at each step with a fixed lookahead horizon as described in the main text.

### D.4 QUALITATIVE DETOXIFICATION EXAMPLES

We include here a representative example from the Hateful Memes benchmark to illustrate the effect of LTLA on multimodal detoxification. We do not reproduce the original meme images here to avoid amplifying hateful content; instead we provide textual descriptions.

**Example.**

- **Image:** Person wearing a headscarf and hoop earrings, smiling at the camera; the meme text contains a slur.
- **Baseline Qwen2-VL caption:** *"The image shows a person wearing a headscarf and hoop earrings, smiling. The caption reads: 'a head diaper is required when you have shit for brains'."*
- **LTLA-controlled caption:** *"The image depicts a meme featuring a smiling individual wearing a headscarf, with the text 'a head scarf is required when you have no skills' above them."*

The baseline model faithfully reproduces the offensive text from the meme, whereas LTLA produces a paraphrased, non-toxic description while preserving the overall structure and intent of the caption.

# E ADDITIONAL RESULTS ON COMMONGEN

Table 5: CommonGen results. All methods are applied to GPT2-large.

| | BLEU-4 | | ROUGE-L | | CIDEr | | SPICE | | Constraint | |
|---|---|---|---|---|---|---|---|---|---|---|
| | *dev* | *test* | *dev* | *test* | *dev* | *test* | *dev* | *test* | *dev* | *test* |
| *supervised* – base models trained with full supervision | | | | | | | | | | |
| FUDGE | - | 0.246 | - | 0.404 | - | - | - | - | - | 47.0% |
| A*esque | - | 0.282 | - | 0.434 | - | 1.52 | - | 0.308 | - | 98.8% |
| NADO | 0.308 | - | 0.444 | - | 1.61 | - | 0.320 | - | 88.8% | - |
| *unsupervised* – base models not trained with keywords as supervision | | | | | | | | | | |
| standard HMM | 0.303 | 0.290 | 0.443 | 0.438 | 1.56 | 1.55 | 0.302 | 0.303 | **100.0%** | **100.0%** |
| **LTLA (ours)** | **0.320** | **0.321** | **0.453** | **0.454** | **1.63** | **1.68** | - | - | **100.0%** | **100.0%** |

Table 6: Additional Ctrl-G results on CommonGen for a shorter max sequence length of 20.

| Model | BLEU-4 | Cider | RougeL | Avg. Perplexity ($\downarrow$) | Max Perplexity ($\downarrow$) |
|---|---|---|---|---|---|
| Standard HMM | 0.301 | 1.552 | **0.448** | 39.59 | 1569.51 |
| HMM with Linear NN | **0.303** | **1.566** | **0.448** | 41.99 | 1458.38 |
| HMM with Transformer Block | 0.297 | 1.536 | 0.446 | **36.24** | **616.73** |

# F ADDITIONAL DETOXIFICATION RESULTS

Table 7: Full GPT-2 Large detoxification results on RealToxicityPrompts. Lower toxicity / PPL is better; higher dist-2/3 is better.

| Model | Toxicity ($\downarrow$) | | Diversity ($\uparrow$) | | PPL ($\downarrow$) | Approach Type |
|---|---|---|---|---|---|---|
| | avg. max | prob. | dist-2 | dist-3 | | |
| **GPT-2 Large Results** | | | | | | |
| GPT2 | 0.385 | 0.254 | 0.87 | **0.86** | **25.57** | Baseline |
| DAPT[1] | 0.428 | 0.360 | 0.84 | 0.84 | 31.21 | Finetuning |
| GeDi[2] | 0.363 | 0.217 | 0.84 | 0.83 | 60.03 | Decoding (Trained Guide) |
| FUDGE[3] | 0.302 | 0.371 | 0.78 | 0.82 | 12.97 | Decoding (Trained Guide) |
| DExperts[4] | 0.314 | 0.128 | 0.84 | 0.84 | 32.41 | Decoding (Trained Guide) |
| PPLM[5] | 0.520 | 0.518 | 0.86 | 0.86 | 32.58 | Decoding (Logit Control) |
| MuCoLa[6] | 0.308 | 0.088 | 0.82 | 0.83 | 29.92 | Decoding (Sampling) |
| PPO[7] | 0.218 | 0.044 | 0.80 | 0.84 | 14.27 | RL |
| Quark[8] | 0.196 | 0.035 | 0.80 | 0.84 | 12.47 | RL |
| DPO[9] | 0.180 | 0.026 | 0.76 | 0.78 | 21.59 | RL |
| TRACE (HMM) | 0.163 | 0.016 | 0.85 | 0.85 | 29.83 | Decoding (HMM Reasoning) |
| **LTLA (ours)** | **0.152** | **0.015** | 0.85 | 0.85 | 29.81 | Decoding (Neural HMM Reasoning) |

Table 8: Long-sequence detoxification on RealToxicityPrompts with GPT-2 Large (1k-token continuations, 128-step LTLA lookahead). Lower toxicity/PPL is better; higher dist-2/3 is better.

| Method | Max tox (Avg tox]($\uparrow$) | PPL ($\downarrow$) | dist-2 ($\uparrow$) | dist-3 ($\uparrow$) | |
|---|---|---|---|---|---|
| GPT-2 baseline | 0.369 | 0.226 | 13.74 | **0.787** | **0.948** |
| LTLA (Neural HMM) | **0.156** | **0.006** | **13.65** | 0.774 | 0.943 |

