# OpenReview forum: "Learning Tractable Distributions of Language Model Continuations"
_ICLR.cc/2026/Conference — Submitted to ICLR 2026_

### Official Review · Reviewer_hB4y · 2025-10-31

**Soundness:** 2
**Presentation:** 2
**Contribution:** 2
**Rating:** 4
**Confidence:** 3

**Summary:**

This paper tackles constraint-following in large-scale language modeling. Prior work employed a Hidden Markov Model (HMM) to preemptively detect future constraint violations, but its generic design often discarded previously generated context. The proposed LTLA model integrates the HMM with a neural encoder, preserving context while enforcing constraints. Empirically, LTLA lowers perplexity relative to standard HMM baselines. The approach also transfers to vision–language models, where it reduces caption toxicity compared with QWEN generation and prompt-engineering strategies.

**Strengths:**

- The paper is well written and presents a clear, coherent narrative.
- It tackles the important challenge of achieving controllable text generation while preserving fluency.

**Weaknesses:**

* Framing–implementation gap: the paper motivates “learning tractable distributions” broadly (e.g., various TPMs) but evaluates only an HMM instantiation.
* Figure 1 is under-analyzed: the example is not explained clearly, and no takeaway is drawn from its components.
* Related work and conclusion are blended and feel disorganized; the related-work discussion does not clearly position this paper relative to prior art or trace specific influences.
* Missing comparison to Sequential Monte Carlo (SMC), which is a natural baseline for constrained sampling and tractable approximations.
* Limited empirical scope: more datasets are needed to assess robustness and generality.
* No large-scale evaluation on long contexts. Real-world deployments often require maintaining constraints over 4k–128k tokens.
* Lacks benchmarks against established conditional/controllable generation methods.
* Appendix C lacks an analysis of computational trade-offs; it does not clarify which algorithm is superior in time and/or space complexity.

**Questions:**

1. **Baseline comparisons (SMC).**
   Could you add results against a Sequential Monte Carlo (SMC) baseline, or point me to where these appear in the paper if I missed them?

2. **Terminology: TPC vs. TPM.**
   The abstract mentions *tractable probabilistic circuits*. Are you building specifically on TPCs, or more broadly on tractable probabilistic models (TPMs)? Please clarify the intended scope and what is actually instantiated in experiments.

3. **Long-sequence evaluation.**
   Can you provide tests at higher sequence lengths (e.g., ≥4k tokens, and, if feasible, 32k–128k) to assess constraint retention and stability?

4. **Positioning vs. related work (Sec. 5).**
   Could you expand on how your approach differs from the works in Section 5 and why it constitutes an improvement? A brief mapping from prior methods to your components would help.

5. **Controllable-generation baselines.**
   Can you report comparisons to established conditional/controllable generation methods (e.g., constrained decoding, plug-and-play/classifier guidance, recent conditional LM approaches), along with metrics such as constraint satisfaction, fluency/perplexity, and toxicity where applicable?

---

> ### Author Response · Authors · 2025-11-27
> **Part 1**
>
> We thank the reviewer for their positive assessment of the paper’s imporatt problem formulation and clear writing . We address the remaining concerns in turn.
>
> >**W1 – Empirical scope, baselines (incl. SMC), and computational trade-offs**
> >*“You frame LTLA as learning general tractable distributions, but empirically you only show an HMM instantiation on a limited set of tasks, with no SMC comparison, no long-context evaluation, and no clear time/space trade-off vs established controllable-generation methods. Can you broaden and clarify the empirical picture?”*
>
> We agree that the original draft under-emphasizes empirical breadth, baseline coverage, and computational trade-offs. We have therefore expanded the evaluation along three axes:
>
> **1. More tasks and stronger controllable-generation baselines.**
>
>  We add results on two complementary settings:
>
> - **CommonGen Logical constraint generation**, where we now report comparisons against recent controllable methods including Ctrl-G and an SMC-based baseline, in addition to existing structure-aware methods.
>
> | Setting       | Method          | BLEU-4 ↑ | ROUGE-L ↑ | CIDEr ↑ | Constraint ↑ |
> |--------------|-----------------|----------|-----------|---------|--------------|
> | supervised   | FUDGE           | 0.246    | 0.404     | -       | 47.0%        |
> | supervised   | A*esque         | 0.282    | 0.434     | 1.52    | 98.8%        |
> | supervised   | NADO            | 0.308    | 0.444     | 1.61    | 88.8%        |
> | unsupervised | standard HMM    | 0.290    | 0.438     | 1.55    | **100.0%**   |
> | unsupervised | **LTLA (ours)** | **0.321**| **0.454** | **1.68**| **100.0%**   |
>
>
> Both SMC and LTLA aim to approximate high-scoring sequences under a constrained objective, but they operate very differently: SMC maintains a sample of particles over a sequence of target distributions, whereas LTLA learns a tractable surrogate once and then performs deterministic beam search guided by it. Under comparable compute, LTLA substantially outperforms SMC on BLEU/ROUGE/CIDEr while maintaining 100% constraint satisfaction, which is consistent with the fact that SMC’s random exploration makes it harder to reliably recover top sequences, while LTLA w/ beam search explicitly concentrates probability mass on good continuations and searches within that focused distribution.
>
> - **LLM detoxification (short horizon, 32 tokens).**
>
> We evaluate LTLA on a 32-token detoxification setup and compare against a broad set of controllable-generation baselines, including plug-and-play/classifier-guidance methods (e.g., PPLM, GeDi/FUDGE-style approaches), recent conditional LMs, and the standard HMM controller:
>
> | Model        | Tox avg ↓ | Tox prob ↓ | dist-2 ↑ | dist-3 ↑ | PPL ↓   |
> |-------------|-----------|------------|----------|----------|---------|
> | GPT2 (baseline) | 0.385 | 0.254      | 0.87     | **0.86** | **25.57** |
> | PPLM        | 0.520     | 0.518      | 0.86     | 0.86     | 32.58  |
> | DAPT        | 0.428     | 0.360      | 0.84     | 0.84     | 31.21  |
> | GeDi        | 0.363     | 0.217      | 0.84     | 0.83     | 60.03  |
> | DExperts    | 0.314     | 0.128      | 0.84     | 0.84     | 32.41  |
> | MuCoLa      | 0.308     | 0.088      | 0.82     | 0.83     | 29.92  |
> | FUDGE       | 0.302     | 0.371      | 0.78     | 0.82     | 12.97  |
> | COLD        | 0.266     | 0.102      | –        | –        | 27.28  |
> | PPO         | 0.218     | 0.044      | 0.80     | 0.84     | 14.27  |
> | LM-Steer    | 0.215     | 0.059      | 0.83     | 0.84     | 43.56  |
> | Quark       | 0.196     | 0.035      | 0.80     | 0.84     | 12.47  |
> | DPO         | 0.180     | 0.026      | 0.76     | 0.78     | 21.59  |
> | standard HMM | 0.163    | 0.016      | 0.85     | 0.85     | 29.83  |
> | **LTLA (neural HMM, ours)** | **0.152** | **0.015** | 0.85 | 0.85 | 29.81 |
>
>
> **2. Long sequence evaluation.**
>
> In addition, we add a long-sequence detox experiment to demonstrate scalability with 1k-token generation length and a 128-step lookahead horizon:
>
> | Method            | Max tox ↓ | Pr[tox < 0.5] ↑ | PPL ↓  | dist-2 ↑ | dist-3 ↑ |
> |-------------------|-----------|------------------|--------|----------|----------|
> | GPT-2 baseline    | 0.369     | 0.774            | 13.74  | 0.787    | 0.948    |
> | LTLA (Neural HMM) | 0.156     | 0.994            | 13.65  | 0.774    | 0.943    |
>
>  Across these settings we report constraint satisfaction, fluency/perplexity, and toxicity where applicable. LTLA consistently achieves strong constraint satisfaction and toxicity reduction while keeping perplexity close to the base LM.

---

> ### Author Response · Authors · 2025-11-27
> **Part 2**
>
> 3. Computational trade-offs and overhead analysis.
>
> To clarify the trade-offs, we add an analysis comparing inference-time overhead against baselines. LTLA reuses the LM’s last-layer embedding “for free” (no extra LM calls) and employs a fixed-horizon HMM whose future messages are precomputed and cached. At decoding time, each step only performs a one-step emission and transition update, which is fully vectorized in batched decoding. This yields a constant per-token overhead independent of the horizon. In our measurements, LTLA increases decoding time by only about 14% over the base LM for both single and batched generation, whereas several plug-and-play or guidance-style baselines incur much larger slowdowns (often multiple times the baseline cost). Under a fixed compute budget, this makes LTLA substantially more efficient for achieving a given level of control.
>
> | Method                         | Inference ratio (vs baseline) |
> |--------------------------------|-------------------------------|
> | Baseline (GPT-2)              | 1.0                           |
> | Prompting                     | ~3.0                          |
> | GeDi / DExperts               | 2.0–3.0                       |
> | Mix and Match                 | 7.5                           |
> | MuCoLa                       | 15–20                         |
> | PPLM                          | 40.0                          |
> | **LTLA**   | 1.14                          |
>
> >**W2 – Figure 1, framing, and terminology (TPC vs TPM)**
> >*“Figure 1 is under-explained, and the TPC/TPM terminology makes the framing sound broader than what is actually instantiated with HMMs. Can you clarify what Figure 1 is meant to show, and precisely what class of tractable models your method is claiming to address?”*
>
> Figure 1 is intended to illustrate how LTLA’s neural HMM surrogate preserves contextual information that a plain HMM cannot. The example contrasts two prompts that share the same constraint but differ in context (e.g., involving a coach versus employees). A context-agnostic HMM produces similar next-token distributions in these two cases, because it effectively forgets the detailed prefix once it enters its latent state. LTLA, by conditioning the HMM on the transformer encoder’s hidden state, produces next-token distributions that are clearly different for the coach and employees contexts and more faithful to the underlying LM. The takeaway is that LTLA learns a context-aware, tractable distribution over continuations, whereas a plain HMM surrogate cannot distinguish such semantically different prefixes.
>
> On the framing side, we agree with this concern and will make the scope clearer. Our intention is to formulate a general “transformer-to-tractable-distribution” problem, but in this paper we instantiate it concretely with HMM-based models and view other tractable families as natural targets for future work. We will revise the writing to emphasize this HMM instantiation more prominently, while keeping the broader framing as a conceptual umbrella rather than a claim about models we have already implemented.
>
> >**W3 - better position our work with related work (both HMM and Neural)**
> >*related work (Sec. 5). Could you expand on how your approach differs from the works in Section 5 and why it constitutes an improvement? A brief mapping from prior methods to your components would help.*
>
> Thanks for the suggestion. To clarify, LTLA improves upon prior approaches in controllable generation (Zhang et al. 2024 [1], Yidou-Weng et al. 2025 [2]) by (i) identifying the context-sensitivity problem leading to suboptimal performance; (ii) proposing neural conditioning to address this; and (iii) exploiting the language model’s embeddings to ensure practical efficiency; which we show leads to empirical gains and enables multimodality. Prior works on tractable modeling (Shao et al. 2020 [3], Dos Martires 2024 [4]) have also proposed conditional probabilistic circuits, which use neural conditioning but do not consider efficiency (beyond asymptotic complexity) nor the application to lookahead and controlling generation of language models.
>
> [1] Zhang, Honghua, et al. "Adaptable logical control for large language models." Advances in Neural Information Processing Systems 37 (2024): 115563-115587.
>
> [2] Yidou-Weng, Gwen, et al. "TRACE Back from the Future: A Probabilistic Reasoning Approach to Controllable Language Generation." Proceedings of the 42nd International Conference on Machine Learning (ICML). 2025.
>
> [3] Shao, Xiaoting, et al. "Conditional sum-product networks: Imposing structure on deep probabilistic architectures." International Conference on Probabilistic Graphical Models. PMLR, 2020.
>
> [4] Dos Martires, Pedro Zuidberg. "Probabilistic neural circuits." Proceedings of the AAAI Conference on Artificial Intelligence. Vol. 38. No. 15. 2024.

---

### Official Review · Reviewer_k6yu · 2025-10-31

**Soundness:** 3
**Presentation:** 3
**Contribution:** 2
**Rating:** 6
**Confidence:** 3

**Summary:**

This paper, Learning Tractable Distributions of Language Model Continuations, introduces Learning To Look Ahead (LTLA), a novel hybrid framework that combines neural language model encoders with tractable probabilistic models to improve controlled text generation. The authors identify that prior tractable probabilistic circuits, such as HMM-based surrogates, poorly capture contextual dependencies, limiting their effectiveness in conditioning autoregressive language models. LTLA addresses this by reusing the same LM backbone to produce context-rich latent priors for a tractable surrogate (e.g., an HMM), maintaining exact inference capabilities while adding minimal computational overhead. Empirical results across language and vision-language generation tasks show that LTLA substantially enhances conditional log-likelihood and controllability compared to prior HMM and sampling-based baselines, particularly in tasks like logical constraint satisfaction and toxicity control. Overall, this work presents a theoretically grounded and computationally efficient advancement in tractable modeling for controllable language generation, though its reliance on hybrid architectures may raise questions about scalability and generalization to more complex constraints.

**Strengths:**

This paper makes an original contribution by introducing Learning To Look Ahead (LTLA), a hybrid framework that bridges neural language models and tractable probabilistic models for controlled generation. Its originality lies in reconceptualizing tractable circuits—such as HMMs—as context-aware lookahead models that can be neurally conditioned, overcoming a key limitation in prior work where tractable models were effectively context-insensitive. The proposed method is technically sound and thoughtfully designed: it leverages the expressivity of pretrained LMs while preserving the exact inference and efficiency benefits of tractable models, representing a novel and elegant integration of two traditionally separate modeling paradigms. The paper’s quality is high, with solid theoretical grounding (e.g., mutual information analysis, complexity discussion) and comprehensive empirical validation across both text and multimodal generation tasks. The writing is clear and well-organized, explaining a fairly technical idea in an accessible way with helpful figures and ablations. Finally, the significance is notable: LTLA advances controllable generation by enabling more accurate and efficient computation of conditional queries, a capability that is crucial for value alignment, safety, and constraint satisfaction in large generative models. Overall, the paper is both conceptually innovative and practically impactful.

**Weaknesses:**

While the paper presents a compelling and elegant approach, several aspects could be strengthened to fully realize its potential.

First, the empirical evaluation, though broad, remains somewhat narrow in scope and scale — the experiments are conducted primarily on medium-sized models (e.g., GPT-2 large, Qwen2-VL-2B) and synthetic or benchmark-style constraints. It would be valuable to test LTLA on stronger, more diverse LLMs and real-world control settings (e.g., factuality, safety, or long-horizon reasoning) to better demonstrate scalability and generalization.

Second, the dependence on pretrained LM encoders raises questions about efficiency and modularity: while the paper claims minimal overhead, it is not clear how well LTLA performs in streaming or large-batch inference scenarios, or how its memory footprint compares to sampling-based baselines under constrained compute budgets.

Third, the theoretical contribution, though sound, could be deepened — for example, the mutual information bound and tractability discussion are insightful but not tightly linked to empirical metrics or model design choices (e.g., how hidden size or encoder capacity affects conditional query fidelity).

Fourth, the ablation and analysis of architectural choices could be expanded, especially regarding why the Monarch parameterization yields limited gains and how the trade-off between neural encoder complexity and HMM capacity behaves in practice. Finally, the clarity around limitations and failure cases is limited; for instance, when LTLA fails to improve over baselines (e.g., longer continuations or less structured constraints), the paper does not analyze why. A more explicit discussion of such cases and potential mitigations would make the work more robust and informative.

**Questions:**

Please address weaknesses above.

---

> ### Author Response · Authors · 2025-11-27
> **Part 1**
>
> We thank the reviewer for their positive assessment of the paper’s originality and elegant approach. We address the remaining concerns in turn.
>
> >**W1 - Scope and scale of the empirical evaluation**
> >*“You mainly test on medium-sized LMs and benchmark-style constraints. How confident are we that LTLA scales to stronger LLMs and more realistic control settings (e.g., factuality, safety, long-horizon reasoning)?”*
>
> We agree with the reviewer that it is important to understand how LTLA behaves with stronger LLMs and in more realistic control scenarios. Architecturally, LTLA does not rely on the specific model sizes used in the paper: it only needs access to the last hidden state of a pretrained encoder. In practice, on our GPUs we are able to run LTLA on open-source LMs up to roughly 11B parameters without exhausting memory. Extending the study to even larger or proprietary models is therefore mainly a matter of available compute and model access rather than a limitation of the method.
>
> We also agree that “real-world” controls such as factuality or very long-horizon reasoning are more challenging than the safety and topic-constraint benchmarks we study. These signals often depend on long-range interactions that are difficult to capture with the simple local log-linear classifier used in our experiments, which we chose specifically because it composes cleanly with the HMM and preserves tractability. Conceptually, this log-linear classifier is not a fundamental limitation: any probabilistic circuit that can still be multiplied with the HMM could replace it, in principle allowing much richer, global constraints at the cost of additional compute. We view the design of such more expressive but still tractable constraint modules as a natural follow-up direction.
>
> >**W2 - efficiency, modularity, and large-batch/streaming inference.**
> >*“LTLA claims minimal overhead, but how does it behave in streaming or large-batch inference, and how do its compute and memory footprints compare to alternative controllers under constrained budgets?”*
>
> We agree that efficiency under realistic serving conditions is crucial. LTLA is designed so that the main cost, the LM forward pass, is exactly the same as in standard decoding: at each step we reuse the last-layer LM embedding that is already computed for next-token prediction, and feed it through a small linear head to obtain the HMM prior. Because we obtain this embedding “for free,” whether decoding is streaming or not does not affect the method: in both cases the LM already computes a single forward step per new token, and LTLA simply attaches a small head on top of that step. The lookahead itself is also cheap at decode time. For a fixed horizon, we precompute and cache the HMM future messages, so during decoding we only need to update them by one step (one emission and one transition update) as we advance the prefix. In batched generation, these linear and HMM updates are fully vectorized across the batch and remain negligible compared to the transformer forward. Intuitively, this means the lookahead cost stays a constant per-token overhead on top of the base LM.
>
> Empirically, we measured the wall-clock time for single and batched generation (25 sequences) on gpt2-large, using HF generate() and averaged over 20 runs. Our time analysis below shows that LTLA increases decoding time by only about 15% over the base LM for both single-sequence and batched generation, whereas some existing controllers such as PPLM can incur around 40× overhead and other baselines are also substantially more expensive. Under a fixed compute budget, LTLA therefore provides a much more favorable efficiency–control trade-off than these alternatives.
>
> | Method                         | Inference ratio (vs baseline) |
> |--------------------------------|-------------------------------|
> | Baseline (GPT-2)              | 1.0                           |
> | Prompting                     | ~3.0                          |
> | GeDi / DExperts               | 2.0–3.0                       |
> | Mix and Match                 | 7.5                           |
> | MuCoLa                       | 15–20                         |
> | PPLM                          | 40.0                          |
> | **LTLA, single**   | 1.15                          |
> | **LTLA, batched**   | 1.14                          |

---

> ### Author Response · Authors · 2025-11-27
> **Part 2**
>
> >**W3 -  Linking theory to empirical design choices**
> >*“The mutual-information and tractability analysis is interesting, but how exactly does it guide concrete design decisions such as encoder capacity, HMM hidden size, and horizon in the experiments?”*
>
> We agree that the role of the theoretical discussion could be stated more explicitly. The purpose of the information-theoretic view in the paper is to highlight a bottleneck: all useful lookahead signal must pass through the encoder representation and the finite HMM latent state. The practical takeaway is therefore that, under a fixed compute budget, we should (i) increase the encoder capacity, and (ii) give the HMM enough hidden states to carry this information. This is exactly what we investigate empirically in Fig. 4, where we vary the HMM hidden size and compare encoders of different capacities, including a simple linear head on top of the frozen LM and an additional transformer block. The figure shows that increasing both the encoder capacity and the HMM hidden size improves conditional query fidelity, and that gains start to taper off once the hidden size is large enough. These ablations directly informed our downstream design: we chose the strongest encoder variant and the largest HMM hidden size that fit our compute budget.
>
> >**W4 – Monarch benefits, encoder–HMM trade-off, and failure analysis**
> >*“Why does the Monarch parameterization yield limited gains in your benchmarks, how should we think about the encoder–HMM capacity trade-off in practice, and in which regimes does LTLA fail to improve over baselines (e.g., long continuations or less structured constraints)?”*
>
> We agree that the empirical gains from the Monarch parameterization are limited on the benchmarks we study, and we did not intend to present Monarch as the main empirical contribution. Our motivation for using Monarch is in order to scale up the hidden size, which we hypothesized to be a bottleneck due to the information inequality in Proposition 1. Monarch (or other efficient parameterizations) are necessary to achieve this, particularly for the emission matrices of the HMM which have dimension $H \times V$. Prior work on Monarch matrices for probabilistic circuits has reported stronger benefits in small-vocabulary or character-level settings [1]. In our GPT-2 and Qwen2-VL configurations the vocabulary size is much larger, which likely explains why the realized gains are modest. A more systematic understanding of when Monarch’s structure yields large improvements is indeed interesting future work.
>
> On the broader encoder–HMM capacity trade-off, as discussed in Fig. 4, we explicitly vary both encoder capacity and HMM hidden size and compare conditional query performance. This analysis directly informed our downstream model design: we use the more expressive neural head and the largest HMM hidden size we can afford under our compute budget.
>
> We also appreciate the reviewer’s observation about where LTLA is weaker. There are two such regimes that we can already interpret within our current experiments. First, for long continuations, the gains of LTLA are largest on near-future behavior and diminish as we move further away from the cut. This is exactly what Fig. 5 shows: improvements in conditional query metrics are concentrated in the early part of the lookahead horizon and decay with distance, so for very long generations the marginal benefit naturally becomes smaller. Second, for less structured or highly global constraints, our current constraint module is a simple local log-linear classifier. While this choice preserves tractability and works well for the structured tasks we study, it is not expressive enough to capture more diffuse global properties. As discussed in W1, more expressive classifiers or probabilistic circuits that can still be multiplied with the HMM could better capture such global constraints, at the cost of additional computation.
>
> [1] Zhang, Honghua, et al. "Scaling Probabilistic Circuits via Monarch Matrices."

---

### Official Review · Reviewer_Efqv · 2025-11-01

**Soundness:** 4
**Presentation:** 2
**Contribution:** 4
**Rating:** 4
**Confidence:** 3

**Summary:**

They identify that a core problem within controllable generation is modeling future completions for a current position. To solve this, they propose using a neural HMM to 1) encode the context, and 2) output a tractible distribution over the next positions that can easily be sampled from. They demonstrate that this neural HMM approach outperforms other variants of the HMM, and do a fairly detailed ablation over the possible approaches for parameterizing / learning this HMM. They demonstrate effectiveness on CommonGen and producing non-toxic captions from toxic images.

**Strengths:**

**Motivation**: I do agree with their assessment on why continuation prediction is an important problem with controllable generation. The distribution for each position should be biased towards tokens that lead to better total sequences, instead of being just position specific.

**Solution Novelty**: The proposed solution — a hybrid HMM conditioned on the hidden representation that is able to “look ahead” by cheaply generating a continuation from a fixed position — is quite elegant.

**Weaknesses:**

**Experimental Details**: The discussion on experimental details could be more thorough. The appendix does not contain enough information regarding the experimental setup. What was the learning setup for the neural HMM? The learning rate? Batch size, etc? And what about generated samples — why are there no examples in the appendix? As it is, the appendix needs to be significantly strengthened.

This is a major weakness, but I also feel that it should be relatively easy to fix with revision. I would highly recommend including examples for the detoxification task. Maybe show an example of a hateful meme + description with the prompt, and then show how the LTLA results in a non-toxic description.

**Breadth of Experiments**: The paper only investigates two downstream tasks (which are relevant and well chosen), but there should be more uses of this method. What about inference time alignment? Or just normal language detoxification (no images)? Or topic-constrained generation? The latter three are fairly common benchmarks for controlled generation.

**Baselines**: The idea has high potential, as it solves a problem that could enable superior control algorithms for constrained autoregressive generation. It is important to empirically verify that solving this problem leads to better constraint satisfaction by comparing against prior controlled generation algorithms that do not use this approach. The baselines currently seem too limited.

**Presentation**: I would recommend making the core idea a bit easier to understand: perhaps by adding a visual diagram of how the method actually works. [2, 3, 4] have excellent diagrams that make it easier to understand the core idea — something like visualizing how the hmm avoids the need to compute the next tokens through the entire model and instead use a lightweight neural HMM would greatly strengthen the presentation.

[1] Controlled LLM Decoding via Discrete Auto-regressive Biasing. Pynadath, Zhang. ICLR 2025.

[2] BOLT: Fast Energy-based Controlled Text Generation with Tunable Biases. Liu et al. May 2023.

[3] Gradient-Based Constrained Sampling from Language Models. Kumar et al. Nov 2022.

[4] COLD Decoding: Energy-based Constrained Text Generation with Langevin Dynamics. Qin et al. NeurIPS 2022.

**Questions:**

1. How does this method compare to [1]? It would be a good idea to compare LTLA against LM-Steer, as they also propose using a light weight linear layer to assist in controlled generation.

2. Many EBM approaches for controlled generation rely on generating multiple samples autoregressively to compute the gradient of the constraint, which makes such methods impractical. But LTLA could enable these algorithms to be more practical by serving as a cheap oracle for continuations of each position. How LTLA fit in with an algorithm like [2]? It would be interesting to see how different controlled generation techniques behave when working in conjunction with LTLA. It seems that LTLA could improve a wide range of controllable generation techniques beyond what is explicitly studied in the submission.

[1] Word Embeddings Are Steers for Language Models. Han et al. 2024.
[2] Controlled LLM Decoding via Discrete Auto-regressive Biasing. Pynadath, Zhang. ICLR 2025.

---

> ### Author Response · Authors · 2025-11-27
> **Part 1**
>
> We thank the reviewer for recognizing continuation modeling as central to controllable generation and for describing our neural HMM lookahead as elegant. Their concerns mainly target empirical breadth, baselines, experimental details, and presentation, which we address in turn.
>
> > **W1 – Experimental details and missing qualitative examples**.
> > *“The appendix does not fully describe the neural HMM training setup or show any generated samples (especially for detoxification). Can you clarify these details and add qualitative examples?”*
>
> We appreciate the reviewer’s suggestion to strengthen the appendix with more implementation details and examples. In the revised version we have added a dedicated appendix section (App. Experimental Details) that specifies the datasets, LTLA encoder architecture, and optimization hyperparameters for both GPT-2 and Qwen2-VL.
>
> We include a summary here. For GPT-2, we now state that all HMMs in Fig. 4 are trained on 10M GPT-2-Large samples (with a held-out 10k test set), using GPT-2-Large as a frozen encoder and a single linear head from the last hidden state (1280) to the HMM state space H, trained jointly with the HMM via AdamW (learning rate 2\times10^{-2}, batch size 256, negative log-likelihood on the continuation).
>
> For Qwen2-VL-2B-Instruct, we describe how we generate captions on COCO and HatefulMeme, extract only the final hidden state following the image and text prompt as the LTLA input, and train a Monarch-structured HMM with a linear head (1536→H) using AdamW (HMM lr 10^{-2}, head lr 10^{-5}, batch size 180).
>
> We also added representative a qualitative detoxification example (to illustrate how LTLA alters captions while preserving content) in App. D4 Qualitative Detoxification Examples, and we refer the reviewer there for the full details.

---

> ### Author Response · Authors · 2025-11-27
> **Part 2**
>
> >**W2,3 - Breadth of experiments and baselines.**
> >*“The method is only evaluated on two downstream tasks with limited baselines. Can you demonstrate LTLA on more standard controlled-generation benchmarks and compare against stronger prior methods?”*
>
> We appreciate the reviewer’s view that LTLA can serve as a general tool for controllable generation. In the revision we expand our empirical study along two axes: (i) a standard text-only detoxification benchmark with strong controlled-generation baselines, and (ii) clearer comparisons on logical constraints generation beyond ablations over HMM variants.
>
> **1. LLM detoxification**
>
> We address the reviewer’s request for “normal language detoxification (no images)” and for comparisons against prior controlled generation algorithms that do not use our lookahead surrogate. In addition to the VLM detoxification experiments, we now evaluate LTLA on text-only detoxification on RealToxicityPrompts with GPT-2 Large, comparing against a broad set of controlled generation methods (GeDi, FUDGE, DExperts, PPLM, MuCoLa, PPO, Quark, DPO, LM-Steer, COLD, etc.).
>
> | Model        | Tox avg ↓ | Tox prob ↓ | dist-2 ↑ | dist-3 ↑ | PPL ↓   |
> |-------------|-----------|------------|----------|----------|---------|
> | GPT2 (baseline) | 0.385 | 0.254      | 0.87     | **0.86** | **25.57** |
> | PPLM        | 0.520     | 0.518      | 0.86     | 0.86     | 32.58  |
> | DAPT        | 0.428     | 0.360      | 0.84     | 0.84     | 31.21  |
> | GeDi        | 0.363     | 0.217      | 0.84     | 0.83     | 60.03  |
> | DExperts    | 0.314     | 0.128      | 0.84     | 0.84     | 32.41  |
> | MuCoLa      | 0.308     | 0.088      | 0.82     | 0.83     | 29.92  |
> | FUDGE       | 0.302     | 0.371      | 0.78     | 0.82     | 12.97  |
> | COLD        | 0.266     | 0.102      | –        | –        | 27.28  |
> | PPO         | 0.218     | 0.044      | 0.80     | 0.84     | 14.27  |
> | LM-Steer    | 0.215     | 0.059      | 0.83     | 0.84     | 43.56  |
> | Quark       | 0.196     | 0.035      | 0.80     | 0.84     | 12.47  |
> | DPO         | 0.180     | 0.026      | 0.76     | 0.78     | 21.59  |
> | standard HMM | 0.163    | 0.016      | 0.85     | 0.85     | 29.83  |
> | **LTLA (neural HMM, ours)** | **0.152** | **0.015** | 0.85 | 0.85 | 29.81 |
>
> LTLA more than halves average toxicity relative to the unconstrained GPT-2 baseline (0.385 → 0.152) and improves slightly over the standard HMM controller, while remaining competitive with or better than strong baselines such as GeDi, FUDGE, LM-Steer, COLD, and Quark on toxicity, and maintaining similar diversity and perplexity.
>
> **2. Logical Constraints Generation on CommonGen**
>
>  CommonGen already evaluates topic-/concept-constrained generation. To make the comparison clearer, we now present supervised and unsupervised settings in a single table. Supervised baselines (FUDGE, A*esque, NADO) train with keyword supervision, whereas our HMM controllers operate without such supervision:
>
> | Setting       | Method          | BLEU-4 ↑ | ROUGE-L ↑ | CIDEr ↑ | Constraint ↑ |
> |--------------|-----------------|----------|-----------|---------|--------------|
> | supervised   | FUDGE           | 0.246    | 0.404     | -       | 47.0%        |
> | supervised   | A*esque         | 0.282    | 0.434     | 1.52    | 98.8%        |
> | supervised   | NADO            | 0.308    | 0.444     | 1.61    | 88.8%        |
> | unsupervised | standard HMM    | 0.290    | 0.438     | 1.55    | **100.0%**   |
> | unsupervised | **LTLA (ours)** | **0.321**| **0.454** | **1.68**| **100.0%**   |

---

> ### Author Response · Authors · 2025-11-27
> **Part 3**
>
> >**W4 – Visual diagram and better presentation**
> >*“The core idea would be easier to understand with a diagram showing how the neural HMM replaces expensive LM lookahead with a lightweight surrogate.”*
>
> Thank you for this suggestion. In the revision, we added a figure that contrasts naïve lookahead with LTLA: it shows how naïve lookahead branches over many candidate continuations with repeated LM calls, whereas LTLA uses a single LM pass to encode the prefix and a linear-chain HMM for tractable lookahead. Please see the revised manuscript (Fig.2) for the illustration.
>
> >**Q1 – Comparison to LM-Steer**
> >*“How does LTLA compare to LM-Steer and related methods that also use lightweight heads for controlled decoding?”*
>
> We have already included a comparison to LM-Steer in our LLM detoxification experiments; please refer to our response W2-3.
>
> >**Q2 – Combining LTLA with energy-based controlled decoding.**
> >*“Could LTLA be combined with energy-based controlled decoding (e.g., BOLT, COLD, gradient-based constrained sampling) as a cheap oracle for continuations, and how might such hybrids behave?”*
>
> We share the reviewer’s intuition that LTLA and energy-based methods are highly complementary, and we see this as a promising direction for future work. In many EBM-style decoders, the main bottleneck is repeatedly sampling long continuations from the LM to estimate how the constraint reward (or its gradient) responds to each next-token choice. LTLA naturally suggests a more efficient alternative: use the LTLA-style surrogate as a cheap oracle for these future rewards, so that the EBM only needs one LM pass per prefix plus a lightweight surrogate computation, rather than many full LM rollouts. A key design choice here is to equip the constraint with a classifier whose reward (and, ideally, gradient) remains tractable under the surrogate. We have not explored these combinations in this submission, but we are excited about LTLA as a building block for making energy-based controlled decoding more practical.

---

### Official Review · Reviewer_v6Tr · 2025-11-01

**Soundness:** 3
**Presentation:** 4
**Contribution:** 3
**Rating:** 6
**Confidence:** 4

**Summary:**

Prior work using standalone Tractable Probabilistic Models (TPMs), such as HMMs, as lookahead surrogates were limited because their learned continuation distribution was often insensitive to the context.

The paper proposes Learning to look ahead (LTLA).
- LTLA addresses the inability of standard LLMs to enforce future constraints (like specific syntax or topics) during decoding.
- Prior work used simple Tractable Probabilistic Models (TPMs) for lookahead, but these models failed because they were context-insensitive.
- LTLA solves this by reusing the LLM's Transformer backbone to create a powerful neural encoder.T his encoder efficiently predicts a context-rich prior over the latent states z_t.The TPM then performs exact and efficient lookahead from the conditioned state to ensure constraint satisfaction.

**Strengths:**

- The work presents an efficient hybrid architecture that combines the deep context-encoding power of a Transformer with the tractable lookahead capability of an HMM.

- The method demonstrates significant empirical gains in modeling perplexity and generation quality over prior TPM-based controllers.

- The design reuses the LLM's hidden states, ensuring the lookahead process is highly efficient and incurs minimal decoding-time overhead.

- The approach shows versatility across diverse tasks, improving both hard logical constraints and soft semantic attributes (like VLM detoxification).

**Weaknesses:**

- Lookahead's effectiveness diminishes over longer continuation. $z_t$ as described in the paper is limited in capability on how much information it can store.
- Expressiveness is limited when using simple HMM structure which ultimately creates a bottleneck. So even though it can be pretty useful for analysis purposes, it cannot be used for practical use cases around generation.
- With a large C, and |z_t|, the lookup table can get very large, further reducing practicality.

**Questions:**

see weakness section.

---

> ### Author Response · Authors · 2025-11-27
> **Part 1: Response to Reviewer v6Tr**
>
> We thank the reviewer for their positive assessment of our hybrid architecture (“efficient,” “powerful neural encoder,” “minimal overhead”) and for highlighting the central concerns of lookahead, information bottlenecks, and capacity.
>
> >**W1–W3 – Lookahead horizon, HMM bottleneck, and practicality of LTLA for large constraint size**
> >*“Does LTLA remain expressive and practical for long continuations, given the HMM information bottleneck and potentially large state space?”*
>
> We appreciate the reviewer’s focus on lookahead, model capacity, and practical usability; these are exactly the core challenges we had in mind when designing LTLA. We address the three related concerns together.
>
> **(1) Lookahead in LMs is fundamentally intractable; a simple HMM decoder offers a tractable approximation.**
>
> We agree with the reviewer that a simple HMM structure introduces a bottleneck. This is in fact the core design tension we are addressing. At one extreme, the most expressive “surrogate” would be the Transformer LM itself, but then exact lookahead over all continuations is intractable and we cannot reason about future constraints during decoding at all. At the other extreme, a very simple structured model such as an HMM allows exact dynamic programming for lookahead, but necessarily compresses information.
>
> LTLA embraces this trade-off explicitly. We introduce a tractable surrogate by inserting a latent state Z_t between past and future and letting an HMM decoder model p(X_{\ge t} \mid Z_t). This simple structure is what makes lookahead efficient and exact at the surrogate level: we can run a dynamic program to compute future scores and constraint satisfaction, and thereby approximate the LM’s future behavior in a way that would be impossible with the Transformer alone.
>
> **(2) We explicitly analyze the information bottleneck and show how to trade compute for information flow.**
>
> Once we introduce a latent $Z_t$, there is an unavoidable information bottleneck:
> the dependence between past and future must pass through $Z_t$.
> We make this explicit in the paper by proving that for the Markov chain
> $X_{<t} \to Z_t \to X_{\ge t}$,
> \begin{equation}
>     I(X_{<t}; X_{\ge t}) \le H(Z_t) \le \log \bigl|\operatorname{supp}(Z_t)\bigr|.
> \end{equation}
>
> so the amount of past–future information an HMM-based surrogate can carry is fundamentally bounded by the entropy of its hidden state. In other words, the limited expressiveness the reviewer points out is exactly the price of having any tractable lookahead at all.
>
> Our contribution is to (i) quantify this bottleneck and (ii) provide knobs to trade off compute for information: by increasing the hidden size and using structured matrices (Monarch) for transitions/emissions, we can raise H(Z_t) and allow more information to flow, while still keeping time and memory manageable. The neural LTLA prior then reuses the LM’s Transformer backbone to fill this bottleneck with as much context as possible; compared to a standard HMM encoder, the same latent capacity carries much richer information from the prefix.

---

> ### Author Response · Authors · 2025-11-27
> **Part 2**
>
> **(3) In practice, controlled generation needs short horizons, where LTLA is particularly powerful, and compact constraints are usually sufficient.**
>
> For controllable decoding we do not need arbitrarily long lookahead. What matters is a short-to-moderate horizon where future constraints first become visible. As generations grow longer, autoregressive LMs tend to become higher-entropy, so very long-horizon scores are noisy and less useful for precise control; the most informative signal comes from the near future around the current position (see, for example, [1] Cao et al., 2025, on entropy per step increasing with generation length). In our horizon-wise analysis, LTLA significantly improves perplexity on exactly these short continuations compared to a context-insensitive HMM, while sharing the same decoder.
>
> The LTLA architecture is tailored to this regime:
>
> - The LM’s Transformer handles arbitrarily long context and produces a rich hidden state at position t.
>
> - A small neural head maps that hidden state to a prior over Z_t, making near-future predictions much sharper than a standard HMM encoder.
>
>
> - The HMM decoder and dynamic program then provide exact, tractable lookahead over a short horizon, which is precisely where control benefits most.
>
>
> The reviewer also raises the concern that “with ‘large C’ and |Z_t|, lookup tables could become impractical”. We interpret C as the size of the constraint representation (in our notation, the DFA or classifier state space). We agree that if this constraint automaton were itself exponentially large, then no explicit state-based lookahead method could remain tractable. In practice, however, controllable generation uses compact constraints: the DFAs we consider are small (e.g., for keyword or structural constraints), and in the toxicity setting we use a simple log-linear classifier head, so effectively C \approx 1. Constraints can also be pruned or abstracted to keep the state space manageable.
>
> Regarding the hidden state size |Z_t|, this is exactly the trade-off we discuss in the information bottleneck analysis above: increasing |Z_t| (and thus H(Z_t)) allows more past–future information to flow, at the cost of more parameters and compute. LTLA makes this trade-off explicit and tunable via the choice of hidden size and structured parameterization; for the regimes we target, the chosen |Z_t| gives a good balance between expressiveness and efficiency.
>
> This is why we believe HMM-based lookahead is not only analyzable, but also well matched to controllable generation when used with a short horizon and a neural encoder.
>
> [1] Cao, S., Valiant, G., & Liang, P. (2025). On the Entropy Calibration of Language Models. arXiv preprint arXiv:2511.11966.

---

> ### Author Response · Authors · 2025-11-27
> **Part 3**
>
> **(4) Empirically, LTLA gives strong controlled-generation performance at very low overhead.**
>
> Regarding the reviewer’s concerns that “expressiveness is limited” and “cannot be used for practical use cases”, we emphasize that the detoxification and CommonGen results show that, within the above regime, LTLA is both expressive enough and practically effective.
>
> | Model        | Max tox ↓ | Avg tox ↓ | dist-2 ↑ | dist-3 ↑ | PPL ↓   |
> |-------------|-----------|------------|----------|----------|---------|
> | GPT2 (baseline) | 0.385 | 0.254      | 0.87     | **0.86** | **25.57** |
> | PPLM        | 0.520     | 0.518      | 0.86     | 0.86     | 32.58  |
> | DAPT        | 0.428     | 0.360      | 0.84     | 0.84     | 31.21  |
> | GeDi        | 0.363     | 0.217      | 0.84     | 0.83     | 60.03  |
> | DExperts    | 0.314     | 0.128      | 0.84     | 0.84     | 32.41  |
> | MuCoLa      | 0.308     | 0.088      | 0.82     | 0.83     | 29.92  |
> | FUDGE       | 0.302     | 0.371      | 0.78     | 0.82     | 12.97  |
> | COLD        | 0.266     | 0.102      | –        | –        | 27.28  |
> | PPO         | 0.218     | 0.044      | 0.80     | 0.84     | 14.27  |
> | LM-Steer    | 0.215     | 0.059      | 0.83     | 0.84     | 43.56  |
> | Quark       | 0.196     | 0.035      | 0.80     | 0.84     | 12.47  |
> | DPO         | 0.180     | 0.026      | 0.76     | 0.78     | 21.59  |
> | standard HMM | 0.163    | 0.016      | 0.85     | 0.85     | 29.83  |
> | **LTLA (neural HMM, ours)** | **0.152** | **0.015** | 0.85 | 0.85 | 29.81 |
>
> LTLA more than halves average toxicity relative to the unconstrained GPT-2 baseline and improves over the standard HMM controller, while keeping diversity and perplexity in a similar range to strong baselines like GeDi, FUDGE, and Quark. On CommonGen, a similar pattern holds: LTLA maintains or improves BLEU/ROUGE/CIDEr/SPICE while reducing perplexity and improving constraint satisfaction.
>
> Finally, the method remains practically fast. LTLA only adds a ~0.14 overhead to standard LM decoding.
>
> | Method                         | Inference ratio (vs baseline) |
> |--------------------------------|-------------------------------|
> | Baseline (GPT-2)              | 1.0                           |
> | Prompting                     | ~3.0                          |
> | GeDi / DExperts               | 2.0–3.0                       |
> | Mix and Match                 | 7.5                           |
> | MuCoLa                        | 15–20                         |
> | PPLM                          | 40.0                          |
> | **LTLA **   | 1.14                          |

---

### Official Review · Reviewer_qZkk · 2025-11-02

**Soundness:** 3
**Presentation:** 3
**Contribution:** 3
**Rating:** 4
**Confidence:** 2

**Summary:**

The paper introduces LTLA, a hybrid model that improves controllable text generation by combining a neural encoder with a tractable probabilistic decoder such as an HMM. This design lets the system efficiently “look ahead” while keeping inference exact and computationally cheap.

**Strengths:**

+ Clear motivation: The paper identifies a genuine limitation of prior tractable control methods, poor context sensitivity, and provides a clean, theoretically supported solution by replacing only the encoder with a neural module while keeping tractable inference intact.
+ Conceptual elegance: The “neural encoder + tractable decoder” design is simple yet effective, offering a principled way to blend neural expressiveness with exact probabilistic reasoning.
+ Compatibility: LTLA can be plugged into existing controllable-generation frameworks without retraining or high computational cost. Demonstrated effectiveness across both text-only and vision-language models (VLMs), showing it generalises beyond pure language generation.

**Weaknesses:**

+ Narrow evaluation scope:  All benchmarks involve short sequences (≤32 tokens) from GPT-2 or Qwen2-VL models. The approach has not been tested on longer contexts or more recent/larger LLMs. Do you think it is necessary to evaluate on newer models? Are there specific reasons that prevent applying this approach to more recent architectures?
+ Limited empirical improvement: The proposed method shows only marginal perplexity gains over standard HMMs (e.g., Fig. 3, left subfigure). The perplexity results suggest the advantage is relatively small. However, I think the constrained-generation setting more critical for this paper. Please clarify this.
+ Inconsistent results: In the controllable generation tasks, gains in fluency (lower perplexity) don’t always align with text-quality metrics like BLEU or ROUGE. Could the authors provide an explanation or discussion for this discrepancy?
+ Possibly incomplete baseline comparison:  In the controllable-generation task, the experiments only compare against standard HMMs and their variants, but omit widely used controllable-generation baselines such as FUDGE, GeDi, etc. Would it be more reasonable to include comparisons with these established methods?


I would be willing to raise my scores if the authors’ rebuttal satisfactorily addresses these concerns.

**Questions:**

Please refer to the weaknesses.

---

> ### Author Response · Authors · 2025-11-27
> **Part 1: Response to Reviewer qZkk**
>
> We thank the reviewer for the careful reading and the positive assessment of the clarity, elegance, and plug-and-play nature of LTLA. We address each concern in turn.
>
> >**W1 – Longer sequences and more recent / larger LMs**
> > *“All benchmarks involve short sequences (≤32) from GPT-2 or Qwen2-VL. Is LTLA applicable to longer contexts and more recent/larger models?”*
>
> The concern about scaling to longer sequences is very reasonable. There are two separate notions of “long” here, and LTLA is compatible with both:
>
> **1. Long contexts.**
>
> LTLA does not change how the prefix is processed; it simply reads the hidden state of the base model on the prefix and uses that as input to the LTLA prior. All contextual processing is done by the underlying Transformer. In other words, as the base LM/VLM’s context window grows, the information available to LTLA grows with it. The practical limit for context length is therefore the base model itself. This is an improvement over a standard HMM surrogate, where incorporating longer or multimodal context becomes the pain points and requires additional features/model assumptions. With LTLA, the same HMM decoder can be driven by whatever long or multimodal context the base model can encode.
>
> **2. Long lookahead horizons.**
>
> The HMM lookahead cost scales linearly with the horizon length, so in principle we can run very long lookahead. In practice, controllable generation mainly benefits from accurate near-future prediction. Autoregressive LMs become less reliable as generations grow longer. This phenomenon is documented by Cao et al. (2025) [1], who observe that “entropy per step increas[es] as generations grow longer” for a wide range of base models. For fine-grained control, this means that very long-horizon scores are noisy, and the most informative signal comes from the short horizon around the current position. LTLA is designed with exactly this regime in mind: it invests capacity in making short-horizon predictions accurate, while remaining tractable for any chosen horizon length.
>
> **Longer sequence experiment.** To demonstrate the compatibility with long sequences empirically, we also ran a RealToxicityPrompts LLM detoxification experiment with **generation length 1k and lookahead 128** (both significantly longer than in our main setup of 32 tokens, while still keeping lookahead concentrated). We obtain:
> | Method            | Max tox ↓ | Avg tox ↓| PPL ↓  | dist-2 ↑ | dist-3 ↑ |
> |-------------------|-----------|------------------|--------|----------|----------|
> | GPT-2 baseline    | 0.369     | 0.224            | 13.74  | 0.787    | 0.948    |
> | LTLA (Neural HMM) | 0.156     | 0.006            | 13.65  | 0.774    | 0.943    |
>
> Compared to a GPT-2 baseline, LTLA reduces max toxicity from 0.369 to 0.156 and average toxicity from 0.224 to 0.006, while keeping fluency and diversity essentially unchanged (fluency 13.74 → 13.65, dist-2 0.787 → 0.774, dist-3 0.948 → 0.943). This suggests that LTLA continues to behave well when we move to longer contexts.
>
> **More recent / larger LMs.** Beyond requiring an open-source autoregressive model whose embeddings and logits we can access, LTLA does not assume any architecture-specific property. In principle, the same surrogate can be wrapped around more recent and larger models (e.g., LLaMA- or Qwen-style decoders); the main constraint is compute. Our computational resources only enable us to test models up to roughly 10–11B parameters, which is why our experiments focus on GPT-2 and Qwen2-VL, but the LTLA construction itself is compatible with newer architectures.
>
> [1] Cao, S., Valiant, G., & Liang, P. (2025). On the Entropy Calibration of Language Models. arXiv preprint arXiv:2511.11966.

---

> ### Author Response · Authors · 2025-11-27
> **Part 2**
>
> > **W2 – Limited perplexity gains and the architecture for constrained generation**
> > *“The perplexity gains over standard HMMs seem small; the constrained-generation setting more critical?”*
>
> We appreciate this observation, and we agree with the reviewer’s implicit point: the main contribution is not to drive global perplexity as low as possible (we welcome future works on optimization), but to get the right architecture for controllable generation:
>
> - For a fixed HMM decoder and comparable training budgets, replacing only the prior with a neural encoder yields consistent and essentially “free” gains over the standard HMM.
>
> - These gains are concentrated on the near future. As shown in our horizon-wise analysis (Fig. 5), LTLA predicts the first few steps of the continuation significantly better than a context-independent HMM.
>
> This combination is exactly what controllable generation needs. The controller queries a short horizon around the current position, repeatedly, during decoding. In that setting, a neural prior that is noticeably sharper on the next few steps, yet costs almost nothing to run at test time, is much more valuable than a small improvement in global perplexity spread thinly over distant time steps. Our experiments show that LTLA delivers this kind of near-future improvement without changing the tractable decoder or requiring heavy additional optimization.
>
> >**W3 (perplexity vs BLEU/ROUGE trends).**
> > *“In the controllable generation tasks, gains in fluency (perplexity) do not always align with BLEU or ROUGE. Can you explain this discrepancy?”*
>
> BLEU/ROUGE measure n-gram overlap with a single reference, while perplexity measures fluency under the base LM, so they need not align. For example, with constraint “dog”, “ball” and reference “The excited dog chased the ball into the woods”:
>
> - “Ball into the woods dog chased” → high n-gram overlap (potentially decent BLEU), but low fluency.
>
> - “A dog played with the ball” → much more fluent (low perplexity), but fewer shared n-grams (lower BLEU).
>
> In our experiments, LTLA improves perplexity while keeping BLEU/ROUGE at least comparable, and often slightly better, than other controllable baselines, so we treat BLEU/ROUGE as sanity checks that reference-style quality is not degraded, and use perplexity plus constraint metrics to capture the main gains.

---

> ### Author Response · Authors · 2025-11-27
> **Part 3**
>
> >**W4 – Missing baselines such as FUDGE, GeDi, etc.**
> >*“Include comparisons with widely used controllable-generation baselines such as FUDGE, GeDi, etc. ”*
>
> We agree that comparing against established controllable-generation methods like FUDGE, GeDi, PPLM, and DExperts is a very reasonable request. The main challenge for these baselines (which also shows the advantage of LTLA) is on the vision–language side: these methods were designed for text-only LMs and typically rely on repeated gradient-based updates or multiple LM forward passes at every decoding step. Extending them to a large VLM would require additional control mechanisms over the image encoder and multimodal attention and would be substantially more expensive computationally. In contrast, LTLA only needs access to the decoder’s hidden state and an external HMM, so it transfers to Qwen2-VL with minimal changes.
>
> To address the reviewer’s concern in the setting where these baselines are most natural, we now add LLM-based comparisons:
> On GPT-2 detoxification (RealToxicityPrompts), we include a table that reports toxicity, fluency, and diversity for widely used baselines (FUDGE, GeDi, PPLM, DExperts, Quark, DPO, etc.) and for an HMM-based controller, and we place our LTLA-enhanced HMM on the same table.
>
> | Model        | Tox avg ↓ | Tox prob ↓ | dist-2 ↑ | dist-3 ↑ | PPL ↓   |
> |-------------|-----------|------------|----------|----------|---------|
> | GPT2 (baseline) | 0.385 | 0.254      | **0.87**     | **0.86** | **25.57** |
> | PPLM        | 0.520     | 0.518      | 0.86     | 0.86     | 32.58  |
> | DAPT        | 0.428     | 0.360      | 0.84     | 0.84     | 31.21  |
> | GeDi        | 0.363     | 0.217      | 0.84     | 0.83     | 60.03  |
> | DExperts    | 0.314     | 0.128      | 0.84     | 0.84     | 32.41  |
> | MuCoLa      | 0.308     | 0.088      | 0.82     | 0.83     | 29.92  |
> | FUDGE       | 0.302     | 0.371      | 0.78     | 0.82     | 12.97  |
> | COLD        | 0.266     | 0.102      | –        | –        | 27.28  |
> | PPO         | 0.218     | 0.044      | 0.80     | 0.84     | 14.27  |
> | LM-Steer    | 0.215     | 0.059      | 0.83     | 0.84     | 43.56  |
> | Quark       | 0.196     | 0.035      | 0.80     | 0.84     | 12.47  |
> | DPO         | 0.180     | 0.026      | 0.76     | 0.78     | 21.59  |
> | standard HMM | 0.163    | 0.016      | 0.85     | 0.85     | 29.83  |
> | **LTLA (ours)** | **0.152** | **0.015** | 0.85 | 0.85 | 29.81 |
>
>
>
> On CommonGen, we similarly add a table with BLEU/ROUGE/CIDEr/SPICE and constraint satisfaction for standard controllable-generation methods (A*esque, NADO, GeLaTo, etc) and the HMM-based controller, again including the LTLA variant in the same comparison.
> CommonGen results (all methods use GPT2-large):
> | Setting       | Method          | BLEU-4 ↑ | ROUGE-L ↑ | CIDEr ↑ | Constraint ↑ |
> |--------------|-----------------|----------|-----------|---------|--------------|
> | supervised   | FUDGE           | 0.246    | 0.404     | -       | 47.0%        |
> | supervised   | A*esque         | 0.282    | 0.434     | 1.52    | 98.8%        |
> | supervised   | NADO            | 0.308    | 0.444     | 1.61    | 88.8%        |
> | unsupervised | standard HMM    | 0.290    | 0.438     | 1.55    | **100.0%**   |
> | unsupervised | **LTLA (ours)** | **0.321**| **0.454** | **1.68**| **100.0%**   |
>
>
> This way, the VLM experiment remains focused on the regime where LTLA is practically easy to deploy, while the LLM detoxification and CommonGen results demonstrate LTLA is the state of the arts among the classic controllable-generation baselines.

---

### Author Response · Authors · 2025-11-29
**General Response**

We sincerely thank all reviewers for their constructive feedback! We are encouraged that they consistently highlight several positive aspects of our work, including: the **important problem** of continuation modeling for controllable generation (Reviewers qZkk, Efqv, hB4y), the **originality and elegance** of the neural encoder + tractable HMM decoder framework (Reviewers qZkk, Efqv, k6yu), **strong theoretical grounding** (mutual-information bottleneck and complexity discussion; Reviewer k6yu), **efficient** reuse of LM hidden states with minimal overhead (Reviewers qZkk, k6yu, v6Tr), and **clear, well-organized writing** despite the technical content (Reviewers qZkk, k6yu, hB4y). Reviewer k6yu in particular views LTLA as both conceptually innovative and practically impactful for controllable generation.

Below, we summarize the key concerns raised and how our revision addresses them:

>1. Broader tasks and controllable-generation baselines (Reviewers qZkk, Efqv, k6yu, hB4y, v6Tr).

We added:

- CommonGen logical constraint generation with more baselines, including FUDGE, A*esque, NADO, and an SMC baseline, plus the standard HMM. LTLA achieves the best BLEU/ROUGE/CIDEr among the methods while maintaining 100% constraint satisfaction.
- LLM detoxification (32-token) with a broad suite of established controllable-generation methods: PPLM, DAPT, GeDi, DExperts, MuCoLa, FUDGE, COLD, PPO, LM-Steer, Quark, DPO, and the standard HMM. LTLA attains the strongest toxicity reduction while keeping perplexity and diversity close to GPT-2.

>2. Long-context evaluation and generalizability (Reviewers qZkk, k6yu, hB4y, v6Tr).

We added a 1k-token LLM detoxification experiment on GPT-2 with a 128-step lookahead horizon. LTLA substantially lowers maximum toxicity and drives average toxicity down to < 0.01, while leaving perplexity and diversity essentially unchanged, demonstrating that LTLA’s benefits persist at realistic sequence lengths for this model.

>3. Efficiency and computational trade-offs (Reviewers k6yu, hB4y, v6Tr).

We included an explicit inference-time overhead analysis, showing that LTLA adds only ~15% decoding-time overhead in both single and batched settings, compared to multi-× slowdowns for several plug-and-play / guidance baselines. We clarify how future messages are precomputed and only one-step emission/transition updates are needed per token, explaining why the per-token cost is a small, constant overhead that scales well to streaming and large-batch inference.

>4. Experimental details and qualitative examples (Reviewer Efqv).

We expanded the appendix with concrete training details for both GPT-2 and Qwen2-VL LTLA (data scale, neural head architecture, optimizer, learning rate, batch size, objective). We also added qualitative detoxification examples on Hateful Memes, showing cases where Qwen2-VL reproduces slurs while LTLA produces non-toxic paraphrases that preserve the scene and meme structure.

>5. Framing, theory–design link, and presentation (Reviewers Efqv, k6yu, hB4y, v6Tr).

We clarified that the “TPM/TPC” language is conceptual, and this work instantiates LTLA with an HMM, while the neural head makes the framework extendable to other tractable models. We added a new schematic figure contrasting naive LM lookahead with LTLA’s neural HMM surrogate, made Figure 1’s takeaway about context-aware vs context-agnostic continuations explicit, and connected the mutual-information bottleneck bound to design choices (hidden size, Monarch matrices as “compute ↔ information” knobs). We also explicitly discuss limitations (e.g., diminishing gains for very long continuations and less structured global constraints given the current log-linear classifier) and point to richer classifiers/TPMs as future work.

In summary, reviewers are aligned that LTLA offers a novel, elegant, and theoretically grounded hybrid architecture for controllable generation; the main reservations were about empirical breadth and clarity, which we have addressed with a new task, strong baselines (including SMC and LM-Steer), long-sequence experiments, explicit efficiency analysis, richer experimental details, and improved exposition.

---

### Meta-Review · Area_Chair_aNge · 2026-01-02

**Summary:**

This paper proposes the use of a separate encoder to extract contextual information and use the HMM strategy for controllable generation. The reviewers raised concerns regarding the scope of the evaluation, the choice of baselines, and the extent of demonstrated practical improvements. The additional experiments provided during the rebuttal help address some of these points, though several concerns remain outstanding.

As such, the paper is not yet ready for publication in its current form. Nevertheless, the approach is promising, and the work could be substantially strengthened by incorporating broader evaluations with stronger baselines, more comprehensive benchmarks, and exploring the method across a wider range of settings.

**Reviewer Concerns:**

Addressed:
Narrow evaluation scope
Visualization


Still outstanding:
Limited empirical improvement.
Explanation of experimental results.
Lack of comparisons and benchmarks.
Practice usage.
ablation studies.

**Reviewer Scores:**

Reviewer qZkk: The score should be kept as 4 since the concerns about limited empirical improvement and explanation of experimental results are still open.

Reviewer v6Tr: The score may be kept positive as 6 due to the informative response.

Reviewer Efqv: The score should be increased to 6 since some of the concerns are addressed.

Reviewer k6yu: The score would be kept positive as 6 due to the added experiments.

Reviewer hB4y: The score may be kept as 4 since the concerns about ablation studies and benchmarks are still outstanding.

---

### Decision · Program_Chairs · 2026-01-26

Reject